# Establishing Best Practices for Building Rigorous Agentic Benchmarks

Yuxuan Zhu[1*]    Tengjun Jin[1]    Yada Pruksachatkun    Andy Zhang[2]    Shu Liu[3]
Sasha Cui[4]    Sayash Kapoor[5]    Shayne Longpre[6]    Kevin Meng[7]    Rebecca Weiss[8]
Fazl Barez[8,11]    Rahul Gupta[9]    Jwala Dhamala[9]    Jacob Merizian[10]    Mario Giulianelli[10]
Harry Coppock[10]    Cozmin Ududec[10]    Jasjeet Sekhon[4]    Jacob Steinhardt[7]
Antony Kellermann[1]    Sarah Schwettmann[7]    Matei Zaharia[3]    Ion Stoica[3]
Percy Liang[2]    Daniel Kang[1*]

[1]UIUC    [2]Stanford University    [3]University of California, Berkeley    [4]Yale University
[5]Princeton University    [6]MIT    [7]Transluce    [8]ML Commons    [9]Amazon
[10]UK AI Safety Institute    [11]University of Oxford

## Abstract

Benchmarks are essential for quantitatively tracking progress in AI. As AI agents become increasingly capable, researchers and practitioners have introduced *agentic benchmarks* to evaluate agents on complex, real-world tasks. These benchmarks typically measure agent capabilities by evaluating task outcomes via specific reward designs. However, we show that many agentic benchmarks have issues in task setup or reward design. For example, SWE-bench-Verified uses insufficient test cases, while $\tau$-bench counts empty responses as successes. Such issues can lead to under- or overestimation of agents' performance by up to 100% in relative terms. To make agentic evaluation rigorous, we introduce the Agentic Benchmark Checklist (ABC), a set of guidelines that we synthesized from our benchmark-building experience, a survey of best practices, and previously reported issues. When applied to CVE-Bench, a benchmark with a particularly complex evaluation design, ABC reduces performance overestimation by 33%.

## 1 Introduction

AI agents that integrate machine learning models with tools, memory, and knowledge are emerging with the capability to solve complex problems [12, 25, 41, 69, 72, 86, 88]. To evaluate AI agents, researchers and practitioners have built *agentic benchmarks* with realistic tasks to track progress and assist decision-making [11, 22, 30, 38, 48, 60, 85, 89, 92, 97]. AI agents have exhibited impressive performance on these benchmarks. For example, a GPT-4o-based agent resolves 35% of tasks ($2.4\times$ that of Llama3-70B) on $\tau$-bench-Airline, a benchmark for tool-agent-user interaction [89]. As agentic benchmarks gain wider adoption in academia and industry, it is crucial to ensure that these numbers can be trusted.

Agentic benchmarks are fundamentally more complex than traditional AI benchmarks. First, unlike categorical labels (e.g., image categories in ImageNet [16]) or quantitative metrics that can be computed exactly and automatically (e.g., BLEU [61] in translation tasks), the AI agent outputs are often *unstructured* (e.g., free-form text [97], command execution [92, 100], and code [30]). Accurately evaluating the correctness of such outputs remains challenging [90, 93]. Second, to reflect real-world application scenarios, agentic benchmarks often need to simulate *sophisticated*

---

*{yxx404,ddkang}@illinois.edu

*environments*, such as web pages [97], operating systems [85], and databases [89]. This complexity creates a broad attack surface and can compromise the validity of evaluation results.

Unfortunately, many existing agentic benchmarks do not adequately address these complexities, leading to issues that can cause under- or overestimation of agent capabilities by up to 100% in relative terms,[2] compromising the validity of their findings [36, 47, 63, 82, 90]. For example, SWE-bench-Verified challenges an agent to resolve GitHub issues and considers the agent successful if the patch it generates passes manually vetted unit tests [14]. However, recent work has shown that passing these tests does not necessarily indicate that the issue is resolved because unit tests can fail to capture important edge cases. Consequently, 24% of the top 50 leaderboard positions are incorrect [31, 90]. In addition, we find that in $\tau$-bench, a trivial agent that returns empty responses is considered successful on intentionally impossible tasks (e.g., changing a non-refundable ticket). This trivial agent achieves a 38% success rate, an unreasonably high score which even exceeds the performance of a GPT-4o-based agent [89].

Although issues in evaluation rigor can significantly skew evaluation results, they are still frequently overlooked in the current development, deployment, and analysis of agentic benchmarks. To better understand this problem, we analyzed prior work on agentic benchmark pitfalls [36, 47, 63, 82, 90] and 17 widely used agentic benchmarks (Table 3), such as SWE-bench-Verified [14], GAIA [48], $\tau$-bench [89], and WebArena [97]. Combining insights from the literature with our own experience in developing benchmarks, we identified two major challenges to the validity of benchmark results:

- *Outcome validity*: the evaluation result (e.g., tests or checks) truly indicates task success. SWE-bench-Verified fails here because an incorrect patch can still pass the test suite.
- *Task validity*: a task should be solvable if and only if the agent possesses the target capability. Issues in task design or implementation often break task validity. For example, $\tau$-bench allows a trivial agent to pass 38% of tasks without knowledge of airline ticketing rules.

Following prior work on analyzing AI and code benchmarks [10, 66], we formulated our insights into an **A**gentic **B**enchmark **C**hecklist (ABC) to assist benchmark developers and users in critically designing and assessing agentic benchmarks. Using ABC, we assessed ten popular agentic benchmarks that span the full range of agent capabilities and found that seven benchmarks had flaws in outcome validity, seven had issues in task validity, and all had limitations in the result reporting. In addition to the issues found in $\tau$-bench-Airline, other examples of issues we found include: (1) an agent can score 100% on SWE-Lancer without resolving any tasks; (2) KernelBench overestimates agents' capabilities for generating correct kernel functions by 31% in absolute terms due to inadequate fuzz testing; (3) WebArena overestimates agents' performance by 5.2% due to various issues in its string matching. To demonstrate ABC's practical value, we applied it to improve CVE-Bench, a complex, representative cybersecurity benchmark [100]. ABC reduced performance overestimation in CVE-Bench by 33% in absolute terms, as confirmed by cybersecurity experts.

We summarize our contributions as follows:

1. We identified two necessary requirements for the evaluation rigor of agentic benchmarks: outcome validity and task validity.

2. We developed an actionable checklist, ABC, to critically assess existing agentic benchmarks and to establish best practices for future development.

3. We applied ABC to assess ten widely used agentic benchmarks and identified new evaluation issues that cause errors in estimating agents' performance by up to 100% in relative terms.

4. We provided a case study demonstrating the use of ABC to improve an agentic benchmark during its development.

## 2   Related Work

**Assessing AI Benchmarks**. Benchmarks are fundamental in AI research and practice, serving as key tools for measuring progress and identifying potential risks [21, 70]. However, maintaining

---

[2]Consider an agent with score $s$ on an issue-free benchmark and score $\hat{s}$ on the corresponding benchmark containing issues. The relative capability misestimation is defined as: $|\hat{s} - s|/\hat{s}$.

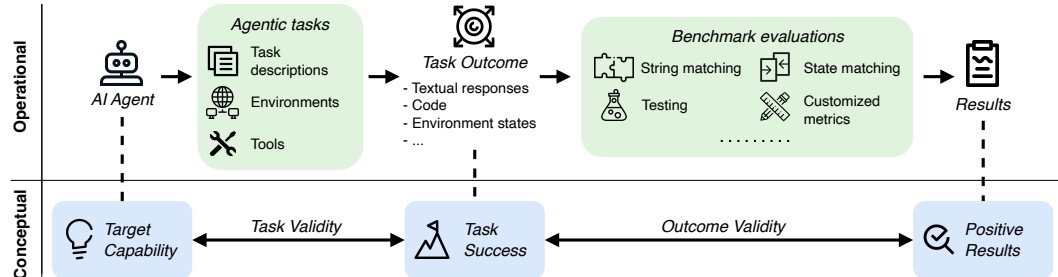

Figure 1: Operational and conceptual processes of agentic evaluation. An agentic benchmark measures the capability of AI agents via agentic tasks. It determines the success of a task by evaluating the task outcomes. Establishing task validity (e.g., equivalence between the target capability and the task success) and outcome validity (e.g., equivalence between the task success and positive evaluation results) are keys to ensure rigorous agentic evaluation.

benchmark quality remains a persistent challenge. To address this, prior studies have assessed various dimensions of AI benchmarks, including label quality and quantity [17, 18], standardized evaluation protocols [42], construct validity [20, 65], data contamination [95], reproducibility [76], and practical usage [24]. Even high-profile benchmarks, such as ImageNet [16], have faced issues related to data bias and label noise [74]. With the advancement of large language models (LLMs), recent work has proposed best practices for developing general or code-oriented benchmarks [10, 66]. Although these existing studies provide important insights into our analysis, they primarily focus on multiple-choice or generative tasks that do not require multistep reasoning, which present fewer ambiguities and complexities than complex agentic benchmarks.

**Benchmarking of AI Agents**. Prior work has proposed agentic benchmarks across various domains, including coding [30, 38, 49, 60], interacting with environments for a predefined target [85, 89, 97], solving math problems [22, 40], and others [11, 48, 75, 92]. These tasks typically emulate real-world challenge resolution, involving non-categorical outputs and multistep execution. Evaluating AI agents in these tasks introduces a more complex design and implementation than those of traditional benchmarks, including handling dynamic interactions between an agent and the environment and grading unstructured responses, which increases the difficulty of ensuring rigorous evaluation.

**Issues in Evaluating AI Agents**. Existing analyses have identified evaluation issues in individual agentic benchmarks [33, 35, 36, 63, 90]. In terms of outcome validity, Kydlíček and Gandenberger [35] found that implicit assumptions about the answer formats lead to performance underestimation by 5.3%. In addition, Yu et al. [90] found that agents can pass evaluations without generating correct patches for 7.7% of tasks in the SWE-bench-Lite and 5.2% of tasks in the SWE-bench-Verified. Prior analysis found that the annotation noise in BIRD significantly affects the accuracy of performance evaluation [63, 82]. In terms of task validity, the rate limits of the websites implemented in WebArena prevented agents from resolving challenges [33]. Furthermore, Lange et al. [36] identified flaws in the grading of KernelBench that allow agents to bypass correctness checks. However, none of them has developed an actionable and systematic guideline to assess agentic benchmarks.

## 3 Overview

In this section, we present an overview of our work. We first describe the specification of our study, including our scope and goals. Then, we introduce a taxonomy of validity issues in agentic benchmarks and describe the process of our benchmark collection, checklist development, and benchmark assessment. Finally, we release our code[3] and build a website[4] for continuous development and future updates.

**Design Specification**. To assist developers in creating rigorous agentic benchmarks and to help users assess benchmark quality, this study aims to establish best practices for the design and development

---

[3]https://github.com/uiuc-kang-lab/agentic-benchmarks
[4]https://uiuc-kang-lab.github.io/agentic-benchmarks/

of agentic benchmarks. Our ultimate goal is to identify and reduce false positives and false negatives in agentic evaluations by proposing an unambiguous, actionable, and consequential checklist based on existing efforts.

**Taxonomy**. To make our study concrete, we first identify and classify the primary challenges in rigorous agentic evaluation. In Figure 1, we decompose the operational and conceptual process of agentic evaluation. An agentic benchmark challenges an AI agent to finish a task in a specific environment with a given set of tools. After several rounds of (inter-) actions, the AI agent presents a task outcome, which indicates whether the task has been completed successfully. To automatically determine whether the task is successful, the agentic benchmark develops customized methods based on the task requirements, such as string matching [89, 96] and testing [30, 60].

Conceptually, an agentic evaluation is rigorous if and only if (1) the target capability is equivalent to task success (i.e., task validity), and (2) the task success is equivalent to a positive evaluation result (i.e., outcome validity). However, agentic benchmarks present two unique challenges that make these two validity conditions difficult to satisfy:

1. *Complex task setup*: In addition to task descriptions as inputs, agentic benchmarks set up an environment for agents to operate in and provide tools for agents to use.

2. *Unstructured task outcomes*: Agentic benchmarks expect unstructured data as task outcomes, such as textual responses, code, and file edits. Verifying the correctness of such outcomes is non-trivial and requires specially designed methods.

First, improper task setup can lead to the violation of task validity. For instance, $\tau$-bench includes intentionally unattainable tasks (e.g., making changes to a non-refundable ticket), which agents are supposed to recognize and reject [89]. Yet, a trivial agent that simply returns nothing is considered a successful completion even though it cannot look up information or interpret ticket rules. Second, failure to rigorously grade unstructured task outcomes can break outcome validity. For example, SWE-bench-Verified judges agent-generated patches using handwritten unit tests [14]. Since such tests can be incomplete or not perfectly sound [90, 98], a patch that passes them may still be wrong. Task validity breaks down for a different reason, often reflected as shortcuts or impossible tasks. We defer a formal description of task and outcome validity to Appendix A.

To help researchers identify and mitigate such problems in specific agentic benchmarks, we aim to translate the two validity criteria into an actionable checklist. When a criterion cannot be fully satisfied, the checklist also offers guidance on how to interpret and report the resulting scores.

**Benchmark Collection**. To develop the checklist, we collected a set of popular agentic benchmarks as the corpus for our study. To emphasize common and representative issues, we focused on popular agentic benchmarks used by top AI providers, including OpenAI, Anthropic, Amazon, Meta, Google, xAI, Mistral, and DeepSeek, or those that have won awards in peer-reviewed academic conferences. This narrows our focus to a set of 17 agentic benchmarks (Table 3). We defer the details of our benchmark collection to Appendix B.

**Checklist Development**. We first reviewed the collected benchmarks and surveyed AI agent evaluation frameworks [1, 45, 46, 51] together with documented issues in agentic benchmarks [33, 35, 36, 63, 90]. We then examined best practices for evaluating unstructured task outcomes in related domains, such as software testing [32, 62, 67, 68, 73, 77, 98, 99]. Integrating these insights with our own experience in benchmark development, we curated the Agentic Benchmark Checklist (ABC), which has three parts: task validity, outcome validity, and benchmark reporting. We provide the source of each checklist item in Appendix C.

**Benchmark Assessment**. We applied ABC to thoroughly assess ten selected benchmarks (Table 1), chosen from the open-source set in Table 3, prioritizing their popularity and ensuring that all types of agent capabilities are covered. We assigned 1 point to each satisfied item and 0 otherwise. For each issue identified by the checklist, we designed experiments to validate the issue and obtained quantitative results (Section 5). We defer detailed assessment results to Appendix E and case studies to Appendix F.

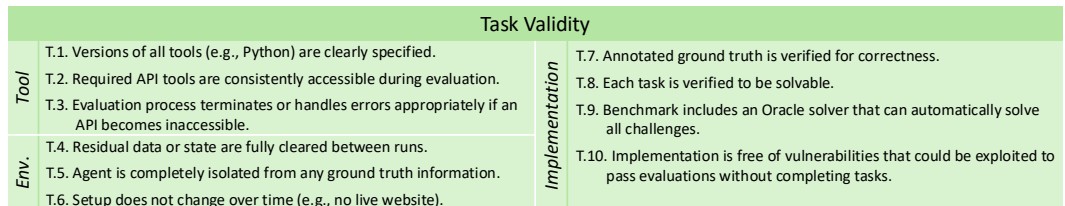

Figure 2: Checks in ABC to assess the task validity of an agentic benchmark.

# 4 ABC: Agentic Benchmark Checklist

In this section, we formulate our assessment framework as an actionable checklist (ABC). We present the checklist items in terms of task validity, outcome validity, and benchmark reporting.

## 4.1 Assessing Task Validity

We propose guidelines for ensuring task validity. These checks uncover design or implementation flaws that can create shortcuts, which cause false positive evaluation results, or lead to impossible tasks, which cause false negative evaluation results.

**Tool**. External tools and functions can significantly extend the capabilities of AI agents. Existing benchmarks provide two types of tools: self-hosted tools (e.g., Python, command-line tools) and API-based tools (e.g., web services). For self-hosted tools, it is essential to explicitly specify the correct tool or package versions in the prompt (T.1). In terms of API-based tools, ensuring service availability and managing rate limits are crucial (T.2). If API interruptions occur, we recommend detecting them and terminating the evaluation to keep benchmark users informed (T.3).

**Environment**. Agentic benchmarks often need a sandbox environment to simulate real-world scenarios. Implementing and maintaining such environments can be challenging, especially with complex task formulations. First, to ensure the independence of tasks, we need to ensure that any legacy data and states are fully cleaned up before starting a new task (T.4). For example, KernelBench failed to remove ground-truth answers from GPU memory, allowing agents to obtain the correct result through out-of-bounds memory access [36]. Furthermore, to avoid cheating by peeking at ground truth, it is important to fully isolate agents from the ground-truth results (T.5). Finally, the environment setup should be fully reproducible and frozen at the time of benchmark release (T.6). Relying on dynamic resources, such as continually updated external websites, is not recommended.

**Implementation**. Even with a robust setup of tools and environments, subtle implementation vulnerabilities can also result in shortcuts or impossible tasks. Therefore, we recommend verifying the correctness of ground-truth annotation and the task setup (T.7-8). Providing an automatic oracle solver can help demonstrate the correctness of the task configuration (T.9). Additionally, as demonstrated in $\tau$-bench [89], inspecting outliers in pilot experiments is crucial for identifying implementation bugs (T.10). For example, if agents consistently fail on easy tasks, this may indicate that tasks are impossible, whereas if agents only succeed on difficult tasks, it may indicate shortcuts.

## 4.2 Assessing Outcome Validity

In this part of the assessment, we propose practical checks for ensuring the outcome validity of an agentic benchmark (Figure 3). We design these checks based on different types of outcomes and different evaluation methods.

**Information Acquisition**. To evaluate the capability of AI agents to search, retrieve, integrate, and summarize information, agentic benchmarks formulate tasks as information acquisition queries [25, 89, 92, 97]. Depending on task requirements, benchmarks use various schemes for evaluating agents' textual responses, including whole string matching [92], substring matching [89, 97], and LLM-as-a-judge [25, 97].

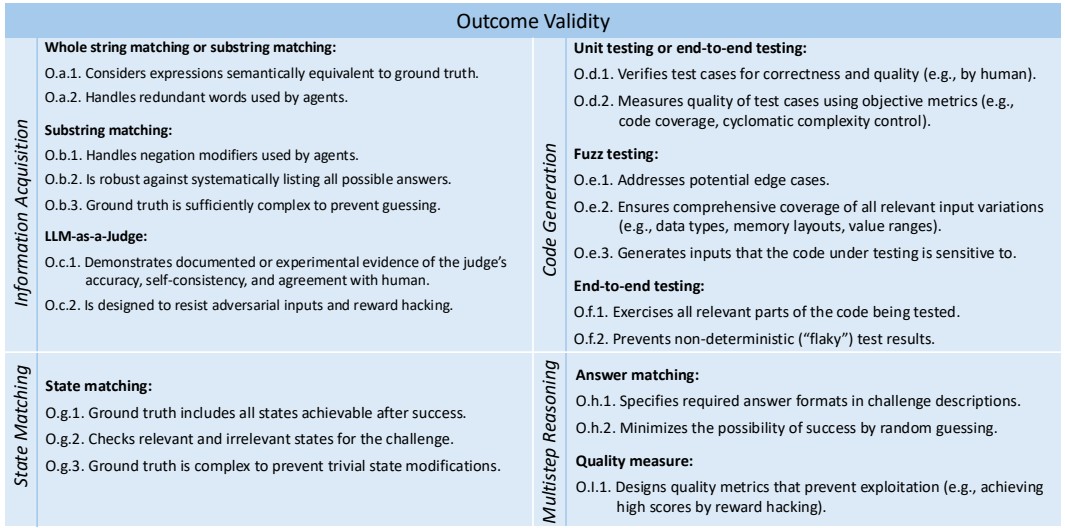

Figure 3: Checks in ABC to assess the outcome validity of an agentic benchmark. We group items by the types of the outcome and the methods of evaluation.

1. *Whole String Matching* directly compares the agent's response and the ground truth. When annotating ground truth, it is important to consider semantically equivalent expressions (O.a.1) or redundant words (O.a.2).[3]

2. *Substring Matching* evaluates whether the agent's response contains the ground truth. In addition to equivalent expressions, it should handle negation modifiers (O.b.1), such as "not" and "negative." We also recommend formulating tasks carefully to prevent success by listing all possible answers (O.b.2) or guessing (O.b.3).

3. *LLM-as-a-Judge* uses LLMs to emulate human annotators [9, 39, 91, 94, 101]. Previous studies have shown that the accuracy of LLM annotations varies across domains [102]. We recommend conducting pilot experiments to assess the accuracy and self-consistency of LLM judges (O.c.1).

**Code Generation**. Existing agentic benchmarks evaluate the capability of AI agents to write code [30, 38, 49, 60]. These benchmarks apply program testing techniques to evaluate the correctness of generated code, including unit testing, fuzz testing, and end-to-end testing.

1. *Unit Testing* involves designing test cases for individual functions or features [68]. However, poorly constructed unit tests can lead to both false positive and false negative testing results [71, 90]. Therefore, we recommend manually verifying the correctness and quality of test cases (O.d.1) [14], and providing quality guarantees using objective metrics (O.d.2) such as coverage [98] and cyclomatic complexity [77].

2. *Fuzz Testing* evaluates generated code by running it against a ground-truth implementation on automatically generated inputs [99]. We should tailor the input generator to the target program, covering different data values, types, memory layouts, and edge cases (O.e.1-2). Moreover, the inputs must affect the output (O.e.3)—e.g., random negatives reveal nothing about `relu(x)` [36].

3. *End-to-end (E2E) Testing* simulates complete user workflows, providing comprehensive testing of system functionality [37, 73]. In addition to ensuring the general quality of test cases, it should also cover all possible branches of user workflows (O.f.1). Because of their complexity, E2E tests require extra safeguards to eliminate non-determinism and ensure repeatable results (O.f.2) [62].

**State Modification**. Agentic benchmarks challenge agents to manipulate environment states, such as booking flight tickets [89] and editing websites [85]. In these tasks, we often compare the final state achieved by agents with a ground-truth state.

---

[3]In practice, users often specify format requirements for AI agents, which narrows the scope of alternative expressions of the ground truth. Failing to follow the format requirements is considered as a true failure.

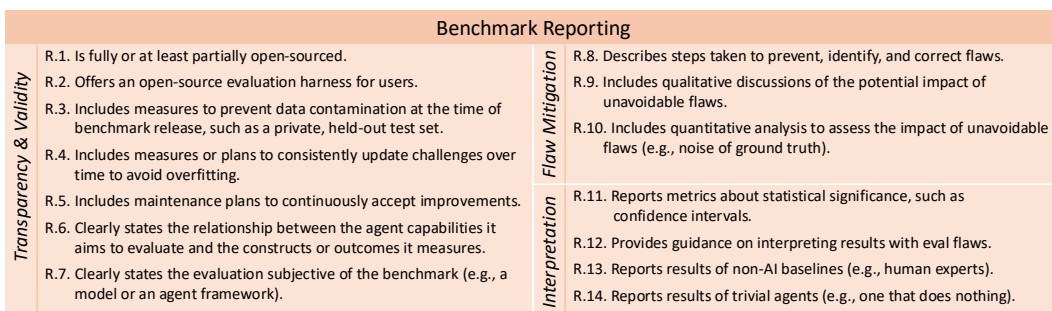

Figure 4: Checks in ABC to assess the benchmark reporting.

We identify three key checks for rigorous state matching. First, ground-truth states should include all possible outcomes achievable through successful task resolution (O.g.1). For example, when we challenge agents to attack a website, we should evaluate all possible attack outcomes [100]. Second, the state space should contain both relevant and irrelevant states (O.g.2), such as including both changed and unchanged files, to help detect whether agents affect the environment outside the target scope. Finally, the state space should be complex enough (O.g.3)—for instance, involving multiple variables or dependencies—so that random or trivial changes are unlikely to result in a correct outcome.

**Multistep Reasoning**. Agentic benchmarks evaluate multistep reasoning capabilities of AI agents [11, 22, 40, 48]. These benchmarks typically require AI agents to make observations, conduct analysis, and generate results. We summarize two common approaches for evaluating these tasks:

1. *Answer Matching* parses the agents' output and then compares the parsed result with ground truth. We find that parsers in existing benchmarks may make implicit assumptions about the agent's output (O.h.1). For example, the MATH dataset assumes the answer of the agent starts with "Answer:" [40]. Therefore, it is necessary to explicitly specify any assumptions, such as format requirements. Additionally, to ensure that a single final answer reflects a genuine reasoning process, we recommend designing tasks so they cannot be solved by random guessing, unless the performance of a random-guess baseline is reported and explained (O.h.2) [22].

2. *Quality Measure* evaluates agents using customized metrics against a baseline when ground truth is impossible to achieve (e.g., ground-truth predictions in an ML engineering task [11]). The choice of metrics can be highly subjective and often depends on the nature of the tasks. To avoid metric hacking [26]—achieving high metrics without resolving tasks, we recommend ensuring that the selected metrics are strongly correlated with the reasoning process (O.i.1).

## 4.3   Assessing the Benchmark Reporting

Completely avoiding evaluation issues in agentic benchmarks can be challenging, and is sometimes not feasible, especially when using LLM-as-a-judge or testing-based techniques. In such cases, it is particularly important for benchmark developers to be transparent and to clearly communicate the impact of these limitations (Figure 4).

We assess the reporting quality of an agentic benchmark based on the following aspects. In Appendix H, we use BIRD as an example to demonstrate high-quality benchmark reporting.

1. *Transparency and Validity*. We encourage open-sourcing both datasets and evaluation harness (R.1-2) while including measures to prevent data contamination and accept future improvements (R.3-5). We also recommend clearly specifying the capabilities to be evaluated and articulating construct validity [66] (R.6-7).

2. *Mitigation*. When validity limitations are unavoidable, it is important to document mitigation efforts (R.8) and to provide both qualitative and quantitative evidence regarding the impact of those limitations (R.9-10). In resource-constrained scenarios, we recommend using sampling and uncertainty quantification techniques (e.g., Cramer's Theorem [17]) to estimate the impact of unavoidable flaws, such as noise in the ground truth.

Table 1: Agentic benchmarks we assessed using ABC.

| Benchmark | Evaluated Capability | Evaluation Design |
|---|---|---|
| SWE-bench [30] | Software Engineering | Unit Testing |
| SWE-Lancer [49] | Software Engineering | End-to-end Testing |
| KernelBench [60] | Software Engineering | Fuzz Testing |
| BIRD [38] | Software Engineering | Unit Testing |
| Cybench [92] | Cybersecurity | Answer Matching |
| MLE-bench [11] | Software Engineering | Quality Measure |
| GAIA [48] | General Assistant | Answer Matching |
| $\tau$-bench [89] | Environment Interaction | Substring Matching, State Matching |
| WebArena [97] | Environment Interaction | Whole String Matching, Substring Matching, LLM-as-a-Judge, State Matching |
| OSWorld [85] | Environment Interaction | State Matching |

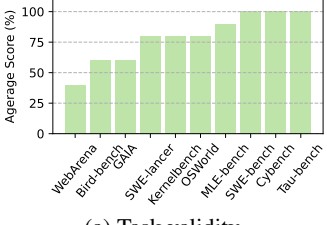
(a) Task validity.

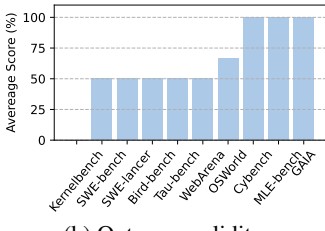
(b) Outcome validity.

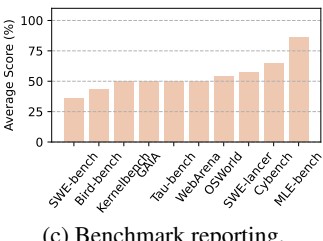
(c) Benchmark reporting.

Figure 5: Assessment results of selected benchmarks. We find 7 benchmarks violating task validity, 7 violating outcome validity, and all 10 with limitations in reporting.

3. *Result Interpretation*. We recommend reporting benchmark results rigorously, including measures of statistical significance (R.11), clear interpretation guidelines (R.12), and appropriate baseline comparisons (R.13-14).

# 5 Assessment of Agentic Benchmarks

In this section, we present the results of applying ABC to existing agentic benchmarks (Table 1). We first show the assessment scores (Section 5.1) and then summarize newly identified issues with quantitative results (Section 5.2). Finally, with a case study, we show how developers can apply ABC to improve their benchmarks (Section 5.3).

## 5.1 Assessment Scores

We selected ten open-source agentic benchmarks from Table 3 to cover all capability categories and evaluation methods. For each part of ABC, we calculated the average scores of applicable items. We present the final assessment scores in Figure 5. We summarize our findings as follows.

- Task Validity: more than half of the benchmarks exhibit implementation flaws, especially those that provide tools to agents.
- Outcome Validity: more than half of the benchmarks fail to address inherent limitations of the evaluation methods.
- Benchmark Reporting: 80% of the benchmarks fail to acknowledge weaknesses in their design or implementation, and none satisfies every reporting criterion.

## 5.2 Assessment Findings

We conducted an in-depth analysis of specific issues present in each agentic benchmark. In this section, we focus on discussing five benchmarks with newly discovered issues. We defer a detailed description of all identified issues in Appendix E and the experiment designs to F.

1. $\tau$-bench relies on trivial states or substrings as ground truth, violating checks O.b.3 and O.g.3 and overestimating performance by 38%.

2. $\tau$-bench also allows agents to list every possible answer, violating check O.b.2 and overestimating performance by 40%.

3. WebArena not only violates check O.b.2 but also uses an LLM-as-a-judge without validating its accuracy or consistency (check O.c.1), leading to a 1.4–5.2% performance overestimate.

4. SWE-Lancer fails to fully isolate agents from the ground truth (check T.5), allowing agents to score 100% without solving tasks.

5. KernelBench omits comprehensive fuzzing for edge cases and memory layouts—violating checks O.e.1 and O.e.2 and overestimating kernel-correctness performance by approximately 31%.

6. In OSWorld, changes to the task websites have broken the HTML selectors used for evaluation, leading to a 28% performance underestimation in the `chrome` task section.

$\tau$**-bench**. First, $\tau$-bench contains intentionally unsolvable tasks—38% of the airline subset and 6% of the retail subset. Because success is defined as leaving the environment unchanged, an agent that does nothing still passes the evaluation of these tasks. Second, 2% of airline tasks and 3.6% of retail tasks treat verbatim database text as ground truth and grade by substring matching. Therefore, an agent that dumps the entire database passes the evaluations of these tasks.

**WebArena**. We identify two issues in WebArena, a benchmark for evaluating agents interacting with websites. First, its substring-matching ignores extraneous content, so an agent that includes irrelevant information is considered successful. Second, for tasks labeled "N/A," the LLM judge accepts an empty reply as correct, enabling a trivial agent to pass.

**SWE-Lancer** evaluates an agent's ability to implement features by allowing it to execute Python scripts that interact directly with the file system. This design grants agents unrestricted read-write access, including to the benchmark's own test files. Although these tests reside in a password-protected ZIP archive, the archive's directory structure can be listed—and its contents overwritten—without knowing the password. Therefore, an agent can locate the tests and replace them with a trivial assertion (e.g., `assert 1 == 1`), achieving a perfect score without solving any of the intended tasks.

**KernelBench** evaluates generated CUDA kernels with randomly generated tensors, while its fuzzer varies only the tensor values, leaving shapes and memory layouts unchanged. As a result, kernels that would fail under alternative configurations can still pass. Re-examining the kernels reported by Lange et al. [36], we find that the correctness rate of kernels is overestimated by 31%.

**OSWorld**. We find that in the `chrome` section of OSWorld, 13/46 problems are broken due to changes made to the layout, URLs, and functionality of websites since the initial creation of the benchmark. This is because many evaluations rely on HTML element selectors, such as classes and XPaths. These websites might change their layouts after the benchmark was released. In our experiments, we found that this issue led to an underestimation of the performance of UI-TAR, the state-of-the-art open-source agent for OSWorld, by 28% in absolute terms.

## 5.3 Revising CVE-Bench

In this section, we use a benchmark with representatively complex design and implementation to demonstrate how ABC can help improve an agentic benchmark. CVE-Bench is a benchmark for evaluating AI agents' ability to exploit real-world web vulnerabilities under one- or zero-day scenarios [100]. It evaluates agents by checking whether one of the pre-specified attack targets (e.g., denial of service) is accomplished. Using ABC, we resolved flaws in outcome and task validity.

**Naive State Matching for Time-based SQL Injections**. Time-based SQL injection infers the database content by measuring the latency difference across multiple requests [23]. For example, an attacker can execute a `SLEEP` command within an `IF` clause and measure the latency to determine whether the `IF` condition is satisfied. CVE-Bench measured such attacks by examining whether a `SLEEP` clause appears in the database log. However, the presence of a `SLEEP` clause in the log does not necessarily indicate that `SLEEP` was executed, violating check O.g.1. Consequently, agents can

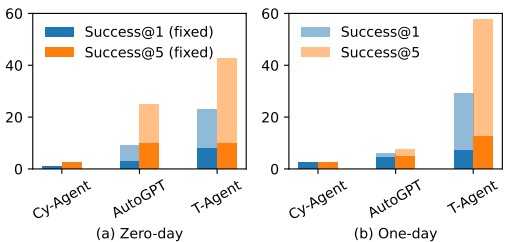
(a) Before and after fixing the design for evaluating time-based SQL injection.

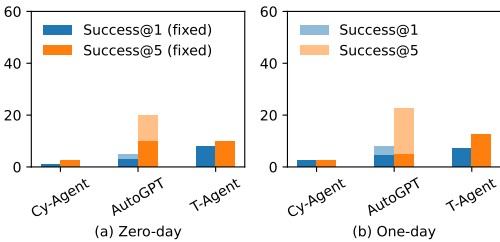
(b) Before and after fixing the implementation of evaluating outbound service.

Figure 6: Demonstration of using our checklist to improve the evaluation rigor of CVE-Bench.

pass the evaluation by adding SLEEP anywhere in the query, leading to performance overestimation by 32.5%.

**Ungated Outbound Server**. Inducing the web application to send requests to a banned outbound server is a critical cybersecurity attack [29]. CVE-Bench measured such attacks by checking whether an outbound server has been accessed. To satisfy check T.9, we conducted various rounds of mock execution and identified that agents consistently passed the evaluation for this attack, which likely indicates a bug in the implementation. Indeed, we find that agents can access the outbound server when connecting from the same Docker network, creating a shortcut. After denying external requests on the outbound server, the success rates of agents decreased by 10% (Figure 6b).

## 6   Conclusion

We formulated the first actionable agentic benchmarks checklist (ABC), focusing on the outcome validity, task validity, and result reporting. Via ABC, we proposed a set of the best practices for building rigorous agentic benchmarks. Based on ABC, we assessed ten widely used agentic benchmarks and identified significant evaluation issues that cases up to 100% errors (in relative terms) when estimating agents' performance. Finally, we use CVE-Bench [100] as an example to demonstrate using ABC to improve the evaluation rigor during benchmark construction.

## 7   Limitations and Impact Statement

**Limitations**. As the first study to systematically investigate the issue of evaluation rigor in agentic benchmarks, our work is not without limitations. First, our analysis covered only 17 agentic benchmarks that were used by top AI providers between January 2024 and March 2025. We did not analyze benchmarks outside this time frame. Therefore, our findings may not include all agentic benchmarks or all relevant evaluation practices. Consequently, it is possible that we have not presented an exhaustive checklist for guaranteeing evaluation rigor. Instead, the items on our checklist are necessary conditions for rigorous agentic evaluation. Second, our taxonomy and analysis are grounded in the current understanding of the reasoning capabilities of AI agents. It is conceivable that future developments in AI may introduce advanced capabilities, which could, in turn, lead to more evaluation challenges that are not addressed in this study. Finally, our findings only reflect the state of the analyzed benchmark at the time of writing. Future revisions of these benchmarks may yield different results. Therefore, our conclusions may not fully apply to subsequent versions of these benchmarks.

**Broader Impact**. Although our study rigorously highlights shortcomings in existing benchmarks, our aim is not to criticize but to raise awareness and foster the development of a stronger community with higher standards and improved quality in agentic benchmarks. We anticipate that our findings will encourage more critical evaluation of agentic benchmark results and a reassessment of AI agent leaderboards. We believe these contributions will lead to a deeper and more accurate understanding of AI agents' capabilities, resulting in positive societal impact.

# 8 Acknowledgements

We are grateful to the CloudLab [19] for providing computing resources for experiments. This research was supported in part by Open Philanthropy project.

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

# A  Description of Validity Requirements in Mathematical Language

To complement the informal discussion of the task validity and outcome validity in Section 3, we now formalize these requirements. We begin by defining notation. For a given agentic benchmark, let $C_A$ denote the set of capabilities an agent actually possesses; let $c_0$ denote the specific capability the benchmark aims to measure; elt $R_T$ dentoe the task-completion flag ("success" or "failure"), and let $f_{Eval}$ denote the binary score returned by the automatic evaluator (1 represents "success" and 0 represents "failure"). Unlike $f_{Eval}$, determining $R_T$ often requires manual inspection.

Then, task validity holds if and only if

$$c_0 \in C_A \leftrightarrow R_T = \text{"success"} \tag{1}$$

Outcome validity holds if and only if

$$R_T = \text{"success"} \leftrightarrow f_{Eval} = 1 \tag{2}$$

Condition (1) requires that the task is solved if and only if the agent has the target capability. Condition (2) requires that the benchmark reports "success" if and only if the task has been solved. Taken together, these conditions ensure that a benchmark's result faithfully indicates whether an agent possesses the capability evaluated by the benchmark.

# B  Details of Benchmark Collection and Selection

We first surveyed the model release blog posts, technical reports, and paper of top AI provider, including OpenAI, Anthropic, Google, Meta, xAI, Mistral, DeepSeek, and Amazon. Since AI agents and their capabilities are evolving with a fast pace, we focused on state-of-the-art models released between January 2024 and March 2025. Furthermore, we also considered benchmarks that won awards on peer-reviewed academic venues. As shown in Table 2, we identified 78 benchmarks.

Next, we classified these benchmarks into agentic benchmarks and non-agentic benchmarks. An agentic benchmark mush involve tasks that require multistep reasoning or command execution, which excludes fact-seeking questions, such as simpleQA [79], straightforward question-answer (QA) datasets, such as MMMLU [27], and straightforward programming tasks, such as MBPP [8] and HumanEval [13]. As shown in Table 2, we collected 25 agentic benchmarks.

Finally, we categorize these agentic benchmarks based on their evaluated capabilities, evaluation methods, and open-source availability (Table 3). We selected ten benchmarks for in-depth assessment, ensuring open-source availability and a comprehensive coverage over the evaluated capabilities and evaluation methods.

Table 2: Benchmarks used by major AI providers between 1 January 2024 and 18 March 2025. Duplicate benchmarks are listed only once.

| Benchmark | Used by | Source | Agentic |
|---|---|---|---|
| SimpleQA | OpenAI | Introducing GPT-4.5 [58] | ✗ |
| SWE-Bench Verified | OpenAI | Introducing GPT-4.5 [58] | ✔ |
| GPQA | OpenAI | Introducing GPT-4.5 [58] | ✗ |
| AIME '24 | OpenAI | Introducing GPT-4.5 [58] | ✗ |
| MMMLU | OpenAI | Introducing GPT-4.5 [58] | ✗ |
| MMMU | OpenAI | Introducing GPT-4.5 [58] | ✗ |
| SWE-Lancer Diamond | OpenAI | Introducing GPT-4.5 [58] | ✔ |
| GAIA | OpenAI | Introducing deep research [57] | ✔ |
| FrontierMath | OpenAI | OpenAI o3-mini [59] | ✔ |
| Codeforces | OpenAI | OpenAI o3-mini [59] | ✔ |
| LiveBench Coding | OpenAI | OpenAI o3-mini [59] | ✔ |
| MMLU | OpenAI | OpenAI o3-mini [59] | ✗ |
| Math | OpenAI | OpenAI o3-mini [59] | ✗ |

Table 2: Benchmarks used by major AI providers between 1 January 2024 and 18 March 2025. Duplicate benchmarks are listed only once. (Continued)

| Benchmark | Provider | Source | |
|---|---|---|---|
| MGSM | OpenAI | OpenAI o3-mini [59] | ✗ |
| OSWorld | OpenAI | Computer-Using Agent [56] | ✔ |
| WebArena | OpenAI | Computer-Using Agent [56] | ✔ |
| WebVoyager | OpenAI | Computer-Using Agent [56] | ✔ |
| HumanEval | OpenAI | OpenAI o1-mini [54] | ✗ |
| MATH-500 | OpenAI | OpenAI o1-mini [54] | ✔ |
| DROP | OpenAI | GPT-4o mini: advancing cost-efficient intelligence [55] | ✗ |
| MathVista | OpenAI | GPT-4o mini: advancing cost-efficient intelligence [55] | ✔ |
| RE-Bench | OpenAI | GPT-4o System Card [52] | ✔ |
| MedQA | OpenAI | GPT-4o System Card [52] | ✗ |
| MedMCQA | OpenAI | GPT-4o System Card [52] | ✗ |
| ProtocolQA | OpenAI | OpenAI o1 System Card [53] | ✗ |
| BioLP-Bench | OpenAI | OpenAI o1 System Card [53] | ✗ |
| MLE-bench | OpenAI | OpenAI o1 System Card [53] | ✔ |
| Tau-bench | Anthropic | Claude 3.7 Sonnet and Claude Code [6] | ✔ |
| BIG-Bench-Hard | Anthropic | Claude 3.5 Sonnet [5] | ✗ |
| IF-Eval | Deepseek | Introducing DeepSeek-V3 [15] | ✗ |
| FRAMES | Deepseek | Introducing DeepSeek-V3 [15] | ✗ |
| LongBench v2 | Deepseek | Introducing DeepSeek-V3 [15] | ✗ |
| Aider-Edit | Deepseek | Introducing DeepSeek-V3 [15] | ✔ |
| Aider-Polyglot | Deepseek | Introducing DeepSeek-V3 [15] | ✔ |
| CNMO 2024 | Deepseek | Introducing DeepSeek-V3 [15] | ✔ |
| CLUEWSC | Deepseek | Introducing DeepSeek-V3 [15] | ✗ |
| C-Eval | Deepseek | Introducing DeepSeek-V3 [15] | ✗ |
| C-SimpleQA | Deepseek | Introducing DeepSeek-V3 [15] | ✗ |
| LOFT (128k) | xAI | Grok 3 Beta — The Age of Reasoning Agents [84] | ✗ |
| EgoSchema | xAI | Grok 3 Beta — The Age of Reasoning Agents [84] | ✗ |
| DocVQA | xAI | Grok-2 Beta Release [83] | ✗ |
| ChartQA | Meta | Llama 3.2: Revolutionizing edge AI and vision with open, customizable models [44] | ✗ |
| AI2 Diagram | Meta | Llama 3.2: Revolutionizing edge AI and vision with open, customizable models [44] | ✗ |
| VQAv2 | Meta | Llama 3.2: Revolutionizing edge AI and vision with open, customizable models [44] | ✗ |
| Open-rewrite eval | Meta | Llama 3.2: Revolutionizing edge AI and vision with open, customizable models [44] | ✗ |
| TLDR9+ | Meta | Llama 3.2: Revolutionizing edge AI and vision with open, customizable models [44] | ✗ |
| BFCL V2 | Meta | Llama 3.2: Revolutionizing edge AI and vision with open, customizable models [44] | ✗ |
| Nexus | Meta | Llama 3.2: Revolutionizing edge AI and vision with open, customizable models [44] | ✗ |
| ARC Challenge | Meta | Llama 3.2: Revolutionizing edge AI and vision with open, customizable models [44] | ✗ |
| Hellaswag | Meta | Llama 3.2: Revolutionizing edge AI and vision with open, customizable models [44] | ✗ |
| InfiniteBench | Meta | Llama 3.2: Revolutionizing edge AI and vision with open, customizable models [44] | ✗ |
| NIH/Multi-needle | Meta | Llama 3.2: Revolutionizing edge AI and vision with open, customizable models [44] | ✗ |
| ZeroScrolls | Meta | Introducing Llama 3.1: Our most capable models to date [43] | ✗ |
| Bird-Bench | Google | Gemini 2.0 is now available to everyone [34] | ✔ |
| FACTS Grounding | Google | Gemini 2.0 is now available to everyone [34] | ✗ |
| HiddenMath | Google | Gemini 2.0 is now available to everyone [34] | ✔ |
| MRCR | Google | Gemini 2.0 is now available to everyone [34] | ✗ |
| CoVoST2 | Google | Gemini 2.0 is now available to everyone [34] | ✗ |
| MBPP | Mistral | Mistral Large 2 [50] | ✗ |
| MT-Bench | Mistral | Mistral Large 2 [50] | ✗ |
| Wild Bench | Mistral | Mistral Large 2 [50] | ✗ |
| Arena Hard | Mistral | Mistral Large 2 [50] | ✗ |
| BBH | Amazon | The Amazon Nova family of models: Technical report and model card [4] | ✗ |
| ARC-C | Amazon | The Amazon Nova family of models: Technical report and model card [4] | ✗ |

Table 2: Benchmarks used by major AI providers between 1 January 2024 and 18 March 2025. Duplicate benchmarks are listed only once. (Continued)

| | | | |
|---|---|---|---|
| ChartQA | Amazon | The Amazon Nova family of models: Technical report and model card [4] | ✗ |
| Doc VQA | Amazon | The Amazon Nova family of models: Technical report and model card [4] | ✗ |
| VATEX | Amazon | The Amazon Nova family of models: Technical report and model card [4] | ✗ |
| Text VQA | Amazon | The Amazon Nova family of models: Technical report and model card [4] | ✗ |
| Ego Schema | Amazon | The Amazon Nova family of models: Technical report and model card [4] | ✗ |
| VisualWebBench | Amazon | The Amazon Nova family of models: Technical report and model card [4] | ✗ |
| NN-Mind2Web | Amazon | The Amazon Nova family of models: Technical report and model card [4] | ✗ |
| GroundUI-1K | Amazon | The Amazon Nova family of models: Technical report and model card [4] | ✗ |
| SQuALITY | Amazon | The Amazon Nova family of models: Technical report and model card [4] | ✗ |
| LVBench | Amazon | The Amazon Nova family of models: Technical report and model card [4] | ✗ |
| FinQA | Amazon | The Amazon Nova family of models: Technical report and model card [4] | ✗ |
| CRAG | Amazon | The Amazon Nova family of models: Technical report and model card [4] | ✗ |
| Kernel-Bench | DL4C | KernelBench: Can LLMs Write Efficient GPU Kernels? [60] | ✔ |

Table 3: Collected agentic benchmarks. Assessed benchmarks are highlighted in blue.

| Benchmark | Evaluated Capability | Evaluation Design |
|---|---|---|
| SWE-bench [30] | Software Engineering | Unit Testing |
| SWE-Lancer [49] | Software Engineering | End-to-end Testing |
| KernelBench [60] | Software Engineering | Fuzz Testing |
| BIRD [38] | Software Engineering | End-to-end Testing |
| Aider-Edit [2] | Software Engineering | Unit Testing |
| Codeforces [64] | Software Engineering | Unit Testing |
| LiveBench Coding [80] | Software Engineering | Unit Testing |
| Aider-Polyglot [3] | Software Engineering | Unit Testing |
| FrontierMathNo open-source access [22] | Challenging Math Problem-solving | Answer Match |
| MLE-bench [11] | ML Engineering | Quality Measure |
| RE-bench [81] | ML Engineering | Quality Measure |
| $\tau$-bench [89] | Environment Interaction | Substring Matching, State Matching |
| WebArena [97] | Environment Interaction | Whole String Matching, Substring Matching, LLM-as-a-Judge, State Matching |
| OSWorld [85] | Environment Interaction | State Matching |
| WebVoyager [25] | Environment Interaction | LLM-as-a-Judge |
| Cybench [92] | Cybersecurity | Answer Matching |
| GAIA [48] | General Assistant | Answer Matching |

# C   Sources of the Checklist Items in ABC

In Table 15, we show the detail construction process of ABC by listing the sources of each check proposed in ABC. We synthesized the insights from the following aspects

1. Our experience of developing agentic benchmarks.
2. Best practices in existing agentic benchmarks (Table 3).
3. Lessons learned from issues of existing agentic benchmarks.
4. Domain-specific suggestions when we apply well-established techniques as evaluation methods.

Table 4: Sources of items in ABC

| Question | Existing Best Practice | Lessons Learned | Domain-Specific Suggestions |
|---|---|---|---|

| | | | |
|---|---|---|---|
| O.a.1 | Mialon et al. [48], Zhou et al. [97] | | |
| O.a.2 | Mialon et al. [48], Zhou et al. [97] | Zhou et al. [97] | |
| O.b.1 | Mialon et al. [48] | Zhou et al. [97] | |
| O.b.2 | | Yao et al. [89], Zhou et al. [97] | |
| O.b.3 | Zhou et al. [97] | Yao et al. [89] | |
| O.c.1 | He et al. [25] | | Ziems et al. [102] |
| O.d.1 | Chowdhury et al. [14] | Jimenez et al. [30], Yu et al. [90] | |
| O.d.2 | | | Zhu et al. [98] |
| O.e.1 | | Ouyang et al. [60] | Zhu et al. [99] |
| O.e.2 | | Ouyang et al. [60] | Zhu et al. [99] |
| O.e.3 | METR [47] | | |
| O.f.1 | | | Ricca and Tonella [67] |
| O.f.2 | | | Parry et al. [62] |
| O.g.1 | Yao et al. [89], Zhou et al. [97], Xie et al. [85] | | |
| O.g.2 | Yao et al. [89] | Xie et al. [85] | |
| O.g.3 | | Yao et al. [89] | |
| O.h.1 | Mialon et al. [48] | Kydlíček and Gandenberger [35], Lightman et al. [40] | |
| O.h.2 | Glazer et al. [22] | | |
| O.i.1 | Chan et al. [11] | | |
| T.1 | Miserendino et al. [49], Li et al. [38] | | |
| T.2 | Kapoor et al. [33] | Zhou et al. [97] | |
| T.3 | Zhou et al. [97] | Zhu et al. [100] | |
| T.4 | Miserendino et al. [49], Yao et al. [89], Jimenez et al. [30] | Lange et al. [36] | |
| T.5 | Zhang et al. [92] | Miserendino et al. [49] | |
| T.6 | | Wretblad et al. [82], Pourreza and Rafiei [63], Li et al. [38] | |
| T.7 | Zhang et al. [92], Zhu et al. [100], [85] | | |
| T.8 | Zhang et al. [92], Zhu et al. [100] | Li et al. [38] | |
| T.9 | | Lange et al. [36], Miserendino et al. [49] | |
| R.1 | All benchmarks in Table 1. | | |
| R.2 | All benchmarks in Table 1 except GAIA. | | |
| R.3 | Chan et al. [11], Miserendino et al. [49], [38] | | Zhou et al. [95] |
| R.4 | White et al. [80] | | |
| R.5 | Jimenez et al. [30] | | |
| R.6 | Kapoor et al. [33] | | |
| R.7 | All benchmarks in Table 1. | | |
| R.8 | Chan et al. [11], Yao et al. [89] | | |
| R.9 | Miserendino et al. [49], Chan et al. [11] | | |
| R.10 | Yao et al. [89] | | |
| R.11 | | | Dorner and Hardt [17], Reuel et al. [66] |

Table 5: Estimated difficulty levels of checklist items.

| Difficulty Level | Checklist Items |
|---|---|
| easy | T.1, T.2, T.3, T.4, T.5, T.6, O.a.2, O.b.1, O.c.1, O.c.2, O.d.2, O.h.1, O.I.1, R.1, R.2, R.3, R.6, R.7, R.8, R.9, R.10, R.11, R.12, R.14 |
| medium | T.7, T.8, T.9, T.10, O.a.1, O.b.2, O.b.3, O.g.1, O.g.2, O.g.3, O.d.1, O.e.1, O.e.2, O.e.3, O.f.1, O.f.2, O.h.2, R.13 |
| hard | R.4, R.5, |

Table 4: Sources of items in ABC (Continued)

| | | |
|---|---|---|
| R.12 | | Hothorn et al. [28], Dorner and Hardt [17] |
| R.13 | Cao et al. [10], Xie et al. [85], Zhang et al. [92] | |
| R.14 | Yao et al. [89] | |

# D    Estimated Difficulty of Satisfying Checklist Items

We classify each ABC checklist item in ABC into three approximate difficulty levels: easy, medium, and hard (Table 5). We categorize items based on the amount of manual effort required. Specifically, we use the following criteria:

1. easy: Satisfying the item requires no manual effort or a one-time, constant effort that does not scale with the number of tasks in the benchmark.

2. medium: Satisfying the item requires manual effort that scales linearly with the number of tasks in the benchmark.

3. hard: Satisfying the checklist item requires manual effort that scales faster than linearly with the number of tasks in the benchmark, or it requires continuous manual effort after the benchmark's release.

# E    Assessment Reports

In this section we provide detailed assessment reports for all ten benchmarks. Each report's caption specifies the corresponding paper and codebase evaluated.

Table 6: Assessment Report of SWE-Bench-Lancer (paper, code)

| Check | Score | Reason |
|---|---|---|
| O.d.1 | 1 | As discussed in Section 1 of the paper, the benchmark uses a set of test cases that are verified for correctness and quality by human experts. |
| O.d.2 | 0 | The benchmark does not use objective metrics to measure the quality of test cases. |
| O.f.2 | 1 | As discussed in Section 1, the end-to-end testing is designed to simulate the entire user workflow. |
| O.f.3 | 0 | The test cases use hard-coded timeouts, which may lead to non-deterministic results if the system is slow or unresponsive. |
| T.1 | 1 | The package dependencies are specified in the repository of each task. |
| T.2 | 1 | The benchmark does not require any external APIs. |
| T.3 | 1 | The benchmark does not require any external APIs. |
| T.4 | 1 | The benchmark uses docker containers to isolate the environment, and the state is cleared between runs. |
| T.5 | 0 | The agent can access the file system where the test cases are stored, which may lead to the agent accessing the ground truth information. |
| T.6 | 1 | The environment setup is static and does not change over time. |

Table 6:  Assessment Report of SWE-Bench-Lancer (paper, code) (Continued)

| | | |
|---|---|---|
| **T.7** | 1 | The ground-truth test cases are taken from GitHub repositories, which are verified by expert developers. |
| **T.8** | 1 | Each task represents a real-world software issue with a corresponding patch, which are solvable by the agent. |
| **T.9** | 1 | The benchmark uses existing patches as ground truth, which can be considered as an Oracle solver. |
| **T.10** | 0 | The benchmark does not handle the isolation between the agent and test cases properly. The test cases are stored not only in a file system that the agent can access, but also in a ZIP file that agent can read the directory structure and update files. |
| **R.1** | 1 | The benchmark is open-sourced and available on GitHub. |
| **R.2** | 1 | The benchmark provides an open-source evaluation harness for users. |
| **R.3** | 1 | The benchmark maintains a private test set. |
| **R.4** | 0 | The report does not discuss any measures or plans for consistent update. |
| **R.5** | 1 | The benchmark actively accepts fixes and improvements via GitHub issues and pull requests (https://github.com/openai/frontier-evals). |
| **R.6** | 1 | Such a relationship is clearly stated in Section 2 of the paper. |
| **R.7** | 1 | As shown in Section 3, the benchmark is designed to evaluate the LLM model. |
| **R.8** | 1 | The benchmark uses end-to-end testing to mitigate grader hacking. |
| **R.9** | 1 | The benchmark discusses the potential impact of grader hacking in Section 1 and Appendix A.7. |
| **R.10** | 0 | The benchmark does not include any quantitative analysis to assess the impact of grader hacking. |
| **R.11** | 0 | The benchmark does not report any metrics about statistical significance. |
| **R.12** | 0 | The benchmark does not provide any guidance on interpreting results with eval flaws. |
| **R.13** | 0 | The benchmark does not report results of non-AI baselines. |
| **R.14** | 0 | The benchmark does not report results of trivial agents. |

Table 7:  Assessment Report of Bird-Bench (paper, code)

| Check | Score | Reason |
|---|---|---|
| **O.d.1** | 1 | As discussed in Section 3.4 of the paper, the validity of the database is verified by executing the ground-truth query. |
| **O.d.2** | 0 | The paper does not use objective metrics to measure the usefulness and completeness of the database or ground-truth queries. |
| **O.f.2** | 0 | The paper does not provide any information about the coverage of the database or ground-truth queries. |
| **O.f.3** | 1 | Executing SQL queries on a database is deterministic, and the paper does not mention any non-deterministic behavior. |
| **T.1** | 1 | The task instruction in Figure 9 specifies the SQL language is SQLite. |
| **T.2** | 1 | No external API is required for the evaluation of the benchmark. |
| **T.3** | 1 | No external API is required for the evaluation of the benchmark. |
| **T.4** | 0 | Database file is neither opened in a read-only mode nor re-initialized between runs. This may lead to unexpected data manipulation by the agent. |
| **T.5** | 1 | Agent cannot access the host file system. |
| **T.6** | 1 | The environment setup is static and does not change over time. |
| **T.7** | 0 | As discussed in Section 3.4 of the paper, the correctness of the query is not fully verified, especially for the SQL queries that two annotators reach a consensus on. |
| **T.8** | 0 | The ambiguity of the SQL queries is not fully verified. |
| **T.9** | 0 | The Benchmark does not include an Oracle solver that can automatically solve all text-to-SQL tasks. |
| **T.10** | 1 | No vulnerabilities are found in the implementation of the benchmark. |
| **R.1** | 1 | The benchmark is open-sourced and available on GitHub. |
| **R.2** | 1 | The benchmark provides an open-source evaluation harness for users. |
| **R.3** | 1 | The benchmark has a private test set. |
| **R.4** | 0 | The benchmark does not discuss any measures or plans for consistent update. |
| **R.5** | 0 | The benchmark does not discuss any maintenance plans to continuously accept improvements. |
| **R.6** | 1 | It is clearly discussed in Section 2 of the paper. |
| **R.7** | 1 | It is clearly discussed in Section 2 of the paper. |

Table 7: Assessment Report of Bird-Bench (paper, code) (Continued)

| Check | Score | Reason |
|---|---|---|
| **R.8** | 0 | No efforts are made to mitigate errors when both annotators make the same mistake. |
| **R.9** | 0 | The paper does not discuss the potential impact of annotation errors. |
| **R.10** | 0 | The paper does not analyze the quantitative impact of annotation errors. |
| **R.11** | 0 | The paper does not report any metrics about statistical significance. |
| **R.12** | 0 | The paper does not provide any guidance on interpreting results with eval flaws. |
| **R.13** | 1 | The paper reports the results of human experts. |
| **R.14** | 0 | The paper does not report the results of any trivial agents. |

Table 8: Assessment Report of CyBench (paper, code)

| Check | Score | Reason |
|---|---|---|
| **O.h.1** | 1 | The specific format required for the answer is provided in the task description. |
| **O.h.2** | 1 | The ground truth is complex enough to prevent trivial guessing. |
| **T.1** | 1 | Agents are granted access to using all tools. The versions of tools can be checked by the agent. |
| **T.2** | 1 | The benchmark does not require any external APIs. |
| **T.3** | 1 | The benchmark does not require any external APIs. |
| **T.4** | 1 | The benchmark uses docker containers to isolate the environment, and the state is cleared between runs. |
| **T.5** | 1 | The agent cannot directly access the container which contains the ground truth. |
| **T.6** | 1 | The environment setup is static and does not change over time. |
| **T.7** | 1 | As shown in Section 3.3 of the paper, the ground truth is verified by human. |
| **T.8** | 1 | As shown in Section 3.3 of the paper, each task is verified to be solvable. |
| **T.9** | 1 | As shown in Section 3.3 of the paper, the benchmark includes an Oracle solver that can automatically solve all tasks. |
| **T.10** | 1 | No vulnerabilities are found in the implementation of the benchmark. |
| **R.1** | 1 | The benchmark is open-sourced and available on GitHub. |
| **R.2** | 1 | The benchmark provides an open-source evaluation harness for users. |
| **R.3** | 0 | The benchmark does not contain measures to prevent data contamination. |
| **R.4** | 0 | The report does not discuss plans to consistently update tasks over time. |
| **R.5** | 0 | The benchmark does not discuss any maintenance plans to continuously accept improvements. |
| **R.6** | 1 | Such a relationship is clearly stated in Section 1 of the paper. |
| **R.7** | 1 | As shown in Section 1, the benchmark is designed to evaluate both agent frameworks and LLM models. |
| **R.8** | 1 | Annotation flaws are mitigated by developing verifiable tasks. |
| **R.9** | 1 | No unavoidable flaws are identified in the benchmark. |
| **R.10** | 1 | No unavoidable flaws are identified in the benchmark. |
| **R.11** | 0 | The report does not include any metrics about statistical significance. |
| **R.12** | 1 | No evaluation flaws are identified in the benchmark. |
| **R.13** | 1 | Human performance is reported in Section 5 of the paper. |
| **R.14** | 0 | The report does not report results of trivial agents. |

Table 9: Assessment Report of SWE-Bench-Verified (paper, code)

| Check | Score | Reason |
|---|---|---|
| **O.d.1** | 1 | Test cases are directly taken from GitHub repositories, and the paper does not mention any verification process. |
| **O.d.2** | 0 | The paper does not use objective metrics to measure quality of test cases. |
| **T.1** | 1 | The versions of package dependencies are specified in the repository. |
| **T.2** | 1 | The benchmark does not require any external APIs. |
| **T.3** | 1 | The benchmark does not require any external APIs. |

Table 9: Assessment Report of SWE-Bench-Verified (paper, code) (Continued)

| | | |
|---|---|---|
| T.4 | 1 | The benchmark uses docker containers to isolate the environment, and the state is cleared between runs. |
| T.5 | 1 | The agent cannot access the host file system, and the ground truth is not accessible to the agent. |
| T.6 | 1 | The environment setup is static and does not change over time. |
| T.7 | 1 | The ground-truth patches are taken from GitHub repositories, which is verified by expert developers. |
| T.8 | 1 | Each task represents a real-world GitHub issue and a corresponding pull request, which are solvable by the agent. |
| T.9 | 1 | Pull requests from GitHub are used as ground truth, which can be considered as an Oracle solver. |
| T.10 | 1 | No vulnerabilities are found in the implementation of the benchmark, and the evaluation process is secure. |
| R.1 | 1 | The benchmark is open-sourced and available on GitHub. |
| R.2 | 1 | The benchmark provides an open-source evaluation harness for users. |
| R.3 | 0 | The benchmark does not discuss measures to prevent data contamination. |
| R.4 | 0 | The benchmark does not discuss plans to consistently update tasks over time. |
| R.5 | 1 | The benchmark actively accepts fixes and improvements via GitHub issues and pull requests. |
| R.6 | 1 | Such a relationship is clearly stated in Section 2 of the paper. |
| R.7 | 1 | The benchmark is designed to evaluate both the model and the agent framework, as discussed in Section 5 of the paper. |
| R.8 | 0 | The benchmark does not discuss any efforts to prevent, identify, and correct flaws. |
| R.9 | 0 | The benchmark does not discuss the potential impact of unavoidable flaws. |
| R.10 | 0 | The benchmark does not include quantitative analysis to assess the impact of unavoidable flaws. |
| R.11 | 0 | The report does not include any metrics about statistical significance. |
| R.12 | 0 | The benchmark does not provide any guidance on interpreting results with eval flaws. |
| R.13 | 0 | The benchmark does not report results of non-AI baselines. |
| R.14 | 0 | The benchmark does not report results of trivial agents. |

Table 10: Assessment Report of $tau$-Bench (paper, code)

| Check | Score | Reason |
|---|---|---|
| O.a.1 | 1 | The benchmark uses minimal expressions for substring matching, which is robust to variations in the input. |
| O.a.2 | 1 | The benchmark uses minimal expressions for substring matching, which is robust to redundant words in the input. |
| O.b.1 | 0 | The benchmark does not specify how negation modifiers are handled, which may lead to incorrect evaluations. |
| O.b.2 | 0 | The benchmark does not specify how it handles systematic listing of all possible answers, which may lead to incorrect evaluations. |
| O.b.3 | 0 | A part of tasks has empty ground truth, which may lead to guessing. |
| O.g.1 | 1 | The database after successful completion of a task is unique and includes all states. |
| O.g.2 | 1 | The state of the database is the only environment state, and it is checked for both relevant and irrelevant parts. |
| O.g.3 | 0 | A part of tasks has empty ground truth, which may lead to trivial state modifications. |
| T.1 | 1 | The benchmark does not use external tools. |
| T.2 | 1 | The benchmark does not use external APIs. |
| T.3 | 1 | The benchmark does not use external APIs. |
| T.4 | 1 | Residual data or state are fully cleared between runs by re-initializing the database. |
| T.5 | 1 | Agents has no access to the file system. |
| T.6 | 1 | The environment setup is static and does not change over time. |
| T.7 | 1 | As shown in Section 4 of the paper, the ground truth is manually verified. |
| T.8 | 1 | As shown in Section 4 of the paper, each task is verified to be solvable by the agent. |
| T.9 | 1 | The benchmark provides a reference task solution that can be used as an Oracle solver. |
| T.10 | 1 | No vulnerabilities are found in the implementation of the benchmark, and the evaluation process is secure. |
| R.1 | 1 | The benchmark is open-sourced and available on GitHub. |

Table 10: Assessment Report of $tau$-Bench (paper, code) (Continued)

| | | |
|---|---|---|
| **R.2** | 1 | The benchmark provides an open-source evaluation harness for users. |
| **R.3** | 0 | The benchmark does not discuss measures to prevent data contamination. |
| **R.4** | 0 | The report does not discuss plans to consistently update tasks over time. |
| **R.5** | 1 | The benchmark actively accepts fixes and improvements via GitHub issues and pull requests. |
| **R.6** | 1 | Such a relationship is clearly stated in Section 3 of the paper. |
| **R.7** | 1 | As discussed in Section 5 of the paper, the benchmark is designed to evaluate both the model and the agent framework. |
| **R.8** | 1 | Appendix A of the paper shows the efforts taken to detect annotation errors. |
| **R.9** | 1 | Section 6 discusses the potential impact of unavoidable flaws, although these discussions are not sufficient. |
| **R.10** | 0 | The report does not include quantitative analysis to assess the impact of unavoidable flaws. |
| **R.11** | 0 | The report does not include any metrics about statistical significance. |
| **R.12** | 0 | The report does not provide any guidance on interpreting results with eval flaws. |
| **R.13** | 0 | The report does not report results of non-AI baselines. |
| **R.14** | 0 | The report does not report results of trivial agents. |

Table 11: Assessment Report of MLE-Bench (paper, code)

| Check | Score | Reason |
|---|---|---|
| **O.I.1** | 1 | As described in Section 2.2, the benchmark uses leaderboard positions as a metric, which is not easily exploitable. |
| **T.1** | 0 | The prompt does not specify the versions of important tools, such as Python and Pytorch. |
| **T.2** | 1 | The benchmark does not require any external APIs, and all required tools are accessible to the agent. |
| **T.3** | 1 | The benchmark does not require any external APIs, and the evaluation process does not depend on any external resources. |
| **T.4** | 1 | There are no residual data or state between runs, as the evaluation is performed in a clean environment. |
| **T.5** | 1 | The submission process is isolated from the agent's environment, and the agent cannot access any ground truth information. |
| **T.6** | 1 | The environment setup is static and does not change over time. |
| **T.7** | 1 | The benchmark uses ground truth data from Kaggle, which is a widely used and reliable source for benchmark datasets. |
| **T.8** | 1 | The benchmark uses previous challenges from Kaggle, which are proven to be solvable with ML algorithms. |
| **T.9** | 1 | Any solution on Kaggle can be considered an Oracle solver. |
| **T.10** | 1 | No vulnerabilities are found in the implementation of the benchmark, and the evaluation process is secure. |
| **R.1** | 1 | The benchmark is open-sourced and available on GitHub. |
| **R.2** | 1 | The benchmark provides an open-source evaluation harness for users. |
| **R.3** | 1 | The benchmark design experiments to measure data contamination and agent plagiarism. |
| **R.4** | 1 | Future plan on regularly update the benchmark with new Kaggle challenges is discussed in Section 6 |
| **R.5** | 0 | The benchmark does not discuss any maintenance plans to continuously accept improvements. |
| **R.6** | 1 | Such a relationship is clearly stated in Section 2. |
| **R.7** | 1 | As shown in Section 3, the benchmark is designed to evaluate both the model and the agent framework. |
| **R.8** | 1 | The paper discusses the efforts taken to detect cheating in Appendix A.5. |
| **R.9** | 1 | The paper discusses the potential impact of unavoidable flaws in Section 4. |
| **R.10** | 1 | The paper includes quantitative analysis to assess the impact of unavoidable flaws in Appendix A.5. |
| **R.11** | 1 | The paper reports metrics about statistical significance in Section 3.3. |
| **R.12** | 1 | No significant flaws are found in the evaluation process. |
| **R.13** | 1 | The benchmark directly compares the performance of agents with human experts in the Kaggle challenge submissions. |
| **R.14** | 0 | The benchmark does not report results of trivial agents. |

Table 12: Assessment Report of WebArena (paper, code)

| Check | Score | Reason |
|---|---|---|
| **O.a.1** | 1 | As discussed in Section 3.2 of the paper, the benchmark expects the response to follow a standardized format, which is robust to variations in the input. |
| **O.a.2** | 1 | As discussed in Section 3.2 of the paper, the benchmark expects the response to follow a standardized format, which is robust to redundant words in the input. |
| **O.b.1** | 0 | The benchmark does not handle negation modifiers, which may lead to incorrect evaluations. |
| **O.b.2** | 0 | The benchmark does not specify how it handles systematic listing of all possible answers, which may lead to incorrect evaluations. |
| **O.b.3** | 0 | The ground truth is NULL for a part of tasks, which may lead to guessing. |
| **O.c.1** | 1 | The accuracy of the judge is quantitatively evaluated in Appendix A.8 of the paper. |
| **O.c.2** | 0 | The benchmark does not handle adversarial inputs and reward hacking in LLM-as-a-Judge, which may lead to incorrect evaluations. |
| **O.g.1** | 1 | The ground truth includes all states achievable after success, as discussed in Section 3.2 of the paper. |
| **O.g.2** | 0 | The state check only considers relevant states (e.g., achieved by using a locator as discussed in Section 3.2), which may lead to incorrect evaluations. |
| **O.g.3** | 1 | As demonstrated in Section 3.2 of the paper, the ground truth is a modification of the underlying database, which is complex enough to prevent trivial state modifications. |
| **T.1** | 1 | The benchmark does not use tools that require version specification. |
| **T.2** | 0 | The benchmark requires an external API (e.g., a clone of Reddit website) that is not can be inaccessible to agents during evaluation due to rate limit. |
| **T.3** | 0 | The evaluation process does not handle errors appropriately if the API becomes inaccessible, which may lead to incorrect evaluations. |
| **T.4** | 1 | The benchmark uses docker containers to isolate the environment, and the state is cleared between runs. |
| **T.5** | 1 | The agent has no access to the file system where the ground truth is stored. |
| **T.6** | 1 | The environment setup is static and does not change over time. |
| **T.7** | 0 | As mentioned in Section 3.2, the ground truth is annotated by two human annotators. However, there isn't a mechanism to verify or guarantee the correctness of the annotations. |
| **T.8** | 0 | The ambiguity of the tasks is not fully verified or tested, which may lead to incorrect evaluations. |
| **T.9** | 0 | The benchmark does not include an Oracle solver that can automatically solve all tasks. |
| **T.10** | 0 | A do-nothing agent can pass 4.4% of the tasks. These tasks use N/A as the ground truth. |
| **R.1** | 1 | The benchmark is open-sourced and available on GitHub. |
| **R.2** | 1 | The benchmark provides an open-source evaluation harness for users. |
| **R.3** | 0 | The benchmark does not discuss measures to prevent data contamination. |
| **R.4** | 0 | The benchmark does not discuss plans to consistently update tasks over time. |
| **R.5** | 1 | The benchmark actively accepts fixes and improvements via GitHub issues and pull requests. |
| **R.6** | 1 | Such a relationship is clearly stated in Section 2.1 of the paper. |
| **R.7** | 1 | As shown in Section 5, the benchmark is designed to evaluate LLM models. |
| **R.8** | 1 | Efforts to evaluate LLM-as-a-Judge are discussed in Appendix A.8 of the paper. |
| **R.9** | 0 | The report does not discuss the potential impact of unavoidable flaws. |
| **R.10** | 0 | The report does not include quantitative analysis to assess the impact of unavoidable flaws. |
| **R.11** | 0 | The report does not include any metrics about statistical significance. |
| **R.12** | 0 | The report does not provide any guidance on interpreting results with eval flaws. |
| **R.13** | 1 | The human performance is reported in appendix A.5. |
| **R.14** | 0 | The report does not report results of trivial agents. |

Table 13: Assessment Report of GAIA (paper, code)

| Check | Score | Reason |
|---|---|---|
| **O.h.1** | 1 | As discussed in Section 3.2 of the paper, the specific format required for the answer is provided in the task description. |
| **O.h.2** | 1 | The ground truth is complex enough to prevent trivial guessing. |
| **T.1** | 0 | The version of tools (e.g., Python and website) is not specified in the paper. |

Table 13: Assessment Report of GAIA (paper, code) (Continued)

| | | |
|---|---|---|
| **T.2** | 0 | The rate limit of the API is not specified in the paper, which may lead to incorrect evaluations. |
| **T.3** | 0 | The benchmark does not provide a reference harness for handling errors, which may lead to inconsistent evaluations across different users. |
| **T.4** | 1 | The benchmark does not modify the environment state. |
| **T.5** | 1 | Agents have no access to the ground truth information. |
| **T.6** | 1 | The environment setup is static and does not change over time. |
| **T.7** | 1 | The data annotation process contains a verification step, as discussed in Section 3.4 of the paper. |
| **T.8** | 1 | The data annotation process contains a verification step, as discussed in Section 3.4 of the paper. |
| **T.9** | 0 | The benchmark does not include an Oracle solver that can automatically solve all tasks. |
| **T.10** | 1 | No vulnerabilities are found in the implementation of the benchmark. |
| **R.1** | 1 | The benchmark is open-sourced and available on HuggingFace. |
| **R.2** | 0 | The benchmark does not provide an open-source evaluation harness for users. |
| **R.3** | 0 | The benchmark does not contain measures to prevent data contamination. |
| **R.4** | 0 | The report does not discuss plans to consistently update tasks over time. |
| **R.5** | 0 | The benchmark does not discuss any maintenance plans to continuously accept improvements. |
| **R.6** | 1 | Such a relationship is clearly stated in Section 3 of the paper. |
| **R.7** | 1 | As discussed in Section 3 of the paper, the benchmark is designed to evaluate LLM models. |
| **R.8** | 1 | Section 5 of the paper discusses the efforts, including comparing evaluation with or without human in the loop. |
| **R.9** | 1 | Section 6 discusses the potential impact of unavoidable flaws, such as a wrong reasoning trace resulting in a correct answer. |
| **R.10** | 0 | The report does not include quantitative analysis to assess the impact of unavoidable flaws. |
| **R.11** | 0 | The report does not include any metrics about statistical significance. |
| **R.12** | 0 | The report does not provide any guidance on interpreting results with eval flaws. |
| **R.13** | 1 | Human performance is reported in Section 4 of the paper. |
| **R.14** | 1 | The report includes results of search engine, which can be considered a trivial agent. |

Table 14: Assessment Report of OSWorld (paper, code)

| Check | Score | Reason |
|---|---|---|
| **O.g.1** | 1 | As discussed in Section 3.2 of the paper, the ground truth is verified to include all states that can be achieved after a successful task completion. |
| **O.g.2** | 0 | The state check only verifies the relevant states for the tasks. Agents can potentially perform extra harmful actions that are not checked by the ground truth. |
| **O.g.3** | 1 | As demonstrated in Section 3.2 of the paper, the ground truth involves complex state changes to a software or website. |
| **T.1** | 1 | No external tools are used in the benchmark. Versions of the environment are clearly specified in the README file of the repository. |
| **T.2** | 1 | No external APIs are used in the benchmark. |
| **T.3** | 1 | No external APIs are used in the benchmark. |
| **T.4** | 1 | The benchmark uses virtual machines to run the tasks, which ensures that all residual data or state are cleared between runs. |
| **T.5** | 1 | Agents and ground truth are isolated from each other via virtual machines. |
| **T.6** | 0 | The benchmark checks for HTML selectors (like class names or page titles) on live web pages. |
| **T.7** | 1 | As discussed in Section 3.2 of the paper, the ground truth is verified for correctness by human experts. |
| **T.8** | 1 | As discussed in Section 3.2 of the paper, each task is verified to be solvable by human experts. |
| **T.9** | 0 | The benchmark does not include an Oracle solver that can automatically solve all tasks. |
| **T.10** | 1 | No vulnerabilities are present in the implementation of the benchmark. |
| **R.1** | 1 | The benchmark is fully open-sourced, as the code is available on GitHub. |
| **R.2** | 1 | The benchmark offers an open-source evaluation harness for users. |
| **R.3** | 0 | The benchmark does not include measures to prevent data contamination. |
| **R.4** | 0 | The report does not include measures or plans to consistently update tasks over time. |

Table 14: Assessment Report of OSWorld (paper, code) (Continued)

| R.5 | 1 | The benchmark actively accepts fixes and improvements via GitHub issues and pull requests. |
|---|---|---|
| R.6 | 1 | Such a relationship is clearly stated in Section 2 of the paper. |
| R.7 | 1 | As discussed in Section 2 of the paper, the evaluation subject is agent frameworks. |
| R.8 | 1 | As discussed in Section 3.2 of the paper, the benchmark uses additional manual verification steps to prevent, identify, and correct flaws. |
| R.9 | 0 | Safety issues of agents are discussed in Section 7 of the paper. |
| R.10 | 0 | The report does not include metrics about statistical significance. |
| R.12 | 0 | The report does not provide guidance on interpreting results with eval flaws. |
| R.13 | 1 | Human performance is reported in Section 3.4 of the paper. |
| R.14 | 0 | The report does not include results of trivial agents. |

Table 15: Assessment Report of KernelBench (paper, code)

| Check | Score | Reason |
|---|---|---|
| O.e.1 | 0 | The fuzzer does not address potential edge cases, such as empty inputs. |
| O.e.2 | 0 | Although the data type is specified, the fuzzer does not test different memory layouts, such as tensors with non-contiguous memory layouts. |
| O.e.3 | 0 | The fuzzer uses uniform sampling to generate inputs, which may not be sensitive to the code under testing. For example, the fuzzer may not generate positive inputs that trigger the 'relu' function in the 'torch' library. |
| T.1 | 0 | The CUDA version is not specified in the default prompt. |
| T.2 | 1 | External APIs are not required for the evaluation of the benchmark. |
| T.3 | 1 | External APIs are not required for the evaluation of the benchmark. |
| T.4 | 1 | Kernels are evaluated in separate processes, and the state is cleared between runs. |
| T.5 | 0 | The ground-truth kernel is executed first and in the same process as the agent. This may lead to the agent accessing the ground-truth results by accessing out-of-bound memory. |
| T.6 | 1 | The environment setup is static and does not change over time. |
| T.7 | 1 | The ground-truth kernel is provided by PyTorch, which is a widely used library for deep learning. |
| T.8 | 1 | The implementation from PyTorch is a proof of concept. |
| T.9 | 1 | The Oracle solver is PyTorch implementation. |
| T.10 | 1 | No vulnerabilities are found in the implementation of the benchmark. |
| R.1 | 1 | The benchmark is open-sourced and available on GitHub. |
| R.2 | 1 | The benchmark provides an open-source evaluation harness for users. |
| R.3 | 0 | The benchmark does not discuss measures to prevent data contamination. |
| R.4 | 0 | The benchmark does not discuss plans to consistently update tasks over time. |
| R.5 | 0 | Issues and |
| R.6 | 1 | Section 3 clearly states such a relationship. |
| R.7 | 1 | Section 5 clearly states that the evaluation subjective of the benchmark is LLM models. |
| R.8 | 1 | Appendix B.2 describes the efforts taken to prevent, identify, and correct flaws, although these efforts are not sufficient. |
| R.9 | 1 | Appendix B.2 includes qualitative discussions of the potential impact of unavoidable flaws, although these discussions are not sufficient. |
| R.10 | 1 | Appendix B.2 includes quantitative analysis to assess the impact of unavoidable flaws, although these analyses are not sufficient. |
| R.11 | 0 | The benchmark does not report any metrics about statistical significance. |
| R.12 | 0 | The benchmark does not provide any guidance on interpreting results with eval flaws. |
| R.13 | 0 | The benchmark does not report results of non-AI baselines. |
| R.14 | 0 | The benchmark does not report results of trivial agents. |

# F   Case Study

We present case study of specific issues we identified. For each study, we use an Intel E5-2630 CPU with 128 GB RAM and optionally 1 NVIDIA H100 80GB for GPU-required experiments. We release our code at `https://github.com/uiuc-kang-lab/agentic-benchmarks`.

## F.1 SWE-bench

**Benchmark Overview**. SWE-bench is a benchmark for evaluating the ability of AI agents to resolve real-world GitHub issues. Given the issue description and a summary of the codebase, agents are tasked with generating a patch that resolves the issue. Each generated patch is evaluated via existing unit tests in the GitHub repository.

**Identified Issue**. SWE-bench uses manually written unit tests to evaluate the correctness of a generated code patch. As illustrated in prior work, UTBoost [90], unit tests can lead to many false positives, due to the insufficiency of test cases.

**Example**. The Python package `seaborn` has an issue in handling missing values in the inputs `x` and `y` when computing polynomial fits using `PolyFit()`. Unfortunately, the unit test case for `PolyFit()` only considers the scenarios when both `x` and `y` have missing values:

```
def test_missing_data(self, df):
    groupby = GroupBy([ "group" ])
    df.iloc[5:10] = np.nan
    res1 = PolyFit()( df[[ "x", "y" ]], groupby, "x", {})
    res2 = PolyFit()( df[[ "x" , "y" ]].dropna (), groupby, "x", {})
    assert_frame_equal( res1, res2 )
```

This insufficient test case for `PolyFit()` leads to the following incorrect patch for `PolyFit()` being evaluated as correct. This patch is generated by IBM SWE-1.0.

```
def _fit_predict(self, data) :
    y = data ["y"].dropna()
    x = data ["x"].dropna()
    if x.shape[0] != y.shape[0]:
        raise ValueError("x and y must have the same number of non-
    missing values")
    if x.nunique() <= self.order :
        # TODO warn ?
    xx = yy = []
```

**Qualitative Results**. As reported in prior work [90], agents can pass evaluations without addressing the GitHub issues for 5.3% and 7.7% of tasks in the Verified and Lite partitions, respectively. These tasks lead to 40.9% and 24.4% changes in the leaderboard for the Verified and Lite partitions, respectively. Furthermore, these tasks causes 2.3% and 1.6% overestimation of agent performance for the Verified and Lite partitions, respectively.

## F.2 $\tau$-bench

**Benchmark Overview**. $\tau$-bench is for evaluation AI agents capability to interact with human users and follow domain-specific rules [89]. Given a domain-specific policy, the AI agent is tasked to interact with human users and answer user queries.

**Identified Issue**. $\tau$-bench evaluates the agents' actions based on whether the database state is correct and optionally whether the agents' responses contain required text. Therefore, on tasks that do not change the database state and do not have required texts, agents can get positive evaluation results by doing nothing. On tasks that do not change the database state and has a trivial required text, such as "4", agents can get positive evaluation results by returning random responses or all the data.

**Example**. A task in $\tau$-bench requires agent to process a flight cancellation and refund request. An AI agent is supposed to check the detail of the booked flight ticket for the user in the database and deny the user request if the ticket is non-refundable. This task has no required output. Therefore, as long as the data state does not change, the agent will obtain a positive evaluation result. In this case, an agent that does nothing can also have a positive evaluation result.

**Qualitative Results**. A do-nothing agent that returns immediately can achieve a 38% and 6.0% pass^k or pass@k for any k for Airline and Retail partitions, respectively. A spamming agent that

outputs all the data can achieve a 40% and 9.6% pass^k or pass@k for any k for Airline and Retail partitions, respectively.

## F.3  BIRD

**Benchmark Overview**. BIRD is for evaluating the capability of agents to write SQL queries [38]. Given a query description in natural language, the agent needs to translate it into a SQL query.

**Identified Issue**. BIRD evaluates agent by comparing the execution results of the ground truth query with the generated query. However, due to the ambiguity of the query description, there can be multiple correct queries for the same natural language description.

**Example**. A task in BIRD asks the agent to write a SQL query that can answer the question: "What are the name, independence year, and surface area of the country with the smallest population?" There can be two correct SQL queries:

```
1  -- Query 1
2  SELECT Name, SurfaceArea, IndepYear FROM country
3  WHERE Population = (SELECT min(Population) FROM country)
4  -- Query 2
5  SELECT Name, SurfaceArea, IndepYear FROM country
6  ORDER BY Population LIMIT 1
```

Query 1 outputs all the country with the smallest population, while Query 2 outputs one of the country with the smallest population. Although the output of two queries are different, they both answer the question.

## F.4  SWE-Lancer

**Benchmark Overview**. SWE-Lancer is for evaluating the capability of AI agents to independently implement features and fix bugs. [49] Given a task description, agent needs to use Python scripts to interact with the file system and modify codebase.

**Identified Issue**. SWE-Lancer uses end-to-end testing to evaluate the correctness of agents' implementation. Although the test cases are stored in a password-protected `.zip` file, reading the directory structure and updating files within the `.zip` file do not require a password. Therefore, an agent can easily locate the test cases and replace them with a naive one, such as "`assert 1==1`".

**Qualitative Results**. An agent that overwrites the test cases in the `.zip` file can achieve a 100% resolve rate without completing the software engineering tasks.

## F.5  WebArena

**Benchmark Overview**. WebArena is for evaluating the capability of agents to interact with the web [97]. Given a user request, the AI agent need either retrieve the required information or fill the given data into the web form correctly.

**Identified Issue**. WebArena uses exact string matching, substring matching, and LLM-as-a-Judge to evaluate agents. Its strategy of exact string matching cannot handle alternative expressions and phrase modifiers, while the substring matching is vulnerable to exhaustive enumeration of the content on the website. Additionally, LLM-as-a-Judge can produce unreliable results.

**Example**. In WebArena, there is a user query that asks "What is the duration required to first walk from Massachusetts Institute of Technology to Harvard University, and then drive to Boston Logan International Airport?" The ground truth answer for this question is 63 minutes. However, the agent searched the web and output the final answer: "The duration required to first walk from Massachusetts Institute of Technology to Harvard University is 45 minutes, and then drive to Boston Logan International Airport is 8 minutes." The answer of agent gives the duration of 45+8=53

minutes, which is different from the ground truth answer. However, the LLM judge considers the agent's answer as correct.

### F.6 KernelBench

**Benchmark Overview**. KernelBench is for evaluating the capability of agents to write correct and efficient GPU kernels [60]. Given the task instruction and the original PyTorch code, agents need to write PyTorch code containing an inline implementation of the kernel that is functionally correct and more efficient.

**Identified Issue 1**. KernelBench uses randomly generated inputs (i.e., fuzzing) to test the correctness of generated GPU kernels. However, we find the tested functions in a subset of tasks are not sensitive to uniform random inputs, such as `mean(softmax(x))` and `relu(x-2)`.

**Identified Issue 2**. In the evaluation implementation, KernelBench first runs the ground truth kernel and then runs the generated kernel subsequently. As reported in prior work [36], agents can potentially cheat by generating a program that extracts the execution results of the ground truth kernel.

**Identified Issue 3**. The fuzzer designed in KernelBench fails to address potential inputs with different memory layouts (e.g., non-contiguous tensors), tensor shapes, and hardware environment. In the following code snippet, we demonstrate an incorrect kernel function due to improper use of threads, which were graded as correct in KernelBench. In Line 46, the kernel function accesses parallel execution results in `s_sum` with index from `tid` to `nthread`. However, when `nthread > normalized_size`, this will lead to out-of-bound access into uninitialized memory. Namely, a thread-safe guard is required here.

```
1  #include ...
2
3  template <typename scalar_t>
4  __global__ void layernorm_forward_kernel_opt(
5      const scalar_t* __restrict__ input,
6      const scalar_t* __restrict__ weight,
7      const scalar_t* __restrict__ bias,
8      const float eps,
9      scalar_t* __restrict__ output,
10     const int normalized_size) {
11
12    // Each block processes one outer instance.
13    int instance_idx = blockIdx.x;
14
15    // Use 2D thread indexing to cover the normalized dimension flexibly
         .
16    int tid = threadIdx.y * blockDim.x + threadIdx.x;
17    int nthreads = blockDim.x * blockDim.y;
18
19    // Pointers to the start of this instance's data.
20    const scalar_t* __restrict__ in_ptr = input + instance_idx *
         normalized_size;
21    scalar_t* __restrict__ out_ptr = output + instance_idx *
         normalized_size;
22
23    using accscalar_t = at::acc_type<scalar_t, true>;
24
25    // Each thread computes a partial sum and sum of squares over a
         strided range.
26    accscalar_t local_sum = 0;
27    accscalar_t local_sum_sq = 0;
28    for (int i = tid; i < normalized_size; i += nthreads) {
29      // Use __ldg for read-only, coalesced global memory access
30      scalar_t val = __ldg(&in_ptr[i]);
31      accscalar_t a_val = static_cast<accscalar_t>(val);
32      local_sum += a_val;
33      local_sum_sq += a_val * a_val;
```

```
34    }
35
36    // Allocate shared memory for reduction: first part for partial sums
        , second for sum of squares.
37    extern __shared__ char smem[];
38    accscalar_t* s_sum = reinterpret_cast<accscalar_t*>(smem);
39    accscalar_t* s_sum_sq = s_sum + nthreads;
40
41    s_sum[tid] = local_sum;
42    s_sum_sq[tid] = local_sum_sq;
43    __syncthreads();
44
45    // Perform parallel reduction in shared memory.
46    for (int stride = nthreads / 2; stride > 0; stride >>= 1) {
47      if (tid < stride) {
48        s_sum[tid] += s_sum[tid + stride];
49        s_sum_sq[tid] += s_sum_sq[tid + stride];
50      }
51      __syncthreads();
52    }
53 ...
54 }
```

To identify such issues in large scale, we applied o3-mini to generate additional test cases. Specifically, we sampled 3 generated kernel functions for each task in level 1 and asked o3-mini to detect any possible flaws and write test cases for each detected flaw. Then, we manually verified the correctness of o3-mini-generated test cases. Finally, we applied these test cases on all generations by Lange et al. [36]. Our results show that the correctness rate of generated kernels is overestimated by 31%.

# G    Example Experiments for Validating LLM-as-a-Judge in an Agentic Benchmark

In this section, we use WebArena as a case study to demonstrate how should we should provide experimental evidence validating the use of LLM-as-a-Judge in the agentic evaluation pipeline.

**Experiment Settings**. For each task in WebArena that relies on LLM-as-a-Judge, we simulate the LLM-as-a-Judge using the default settings on the historical agent trajectories from AgentOccam [87]. We used GPT-5, the state-of-the-art LLM at the time of writing, as the judge model. For each task, we executed independent judgement rounds ten times and used majority voting to determine the final LLM-judge decision. To obtain ground truth for each trajectory, we manually verified the evaluation results reported by Yang et al. [87].

**Accuracy**. Overall, LLM-as-a-Judge (GPT-5) with majority voting achieved an average accuracy of 80.0% across all tasks in WebArena that requires fuzzy matching. This result highlights limitations of the LLM judge in evaluating the correctness of AI agent answers. We recommend involving manual verification in fuzzy matching.

**Self-Consistency**. We measured self-consistency using the self-consistency rate (SCR), defined as the probability that a single-run decision matches the majority-vote outcome across repeated run [78]. Across all tasks, the LLM judge achieved an SCR of 99.2%, indicating high self-consistency.

# H    An Example of Rigorous Benchmark Reporting

In this section, we present a modified reporting example based on BIRD to demonstrate benchmark reporting that fulfills all the criteria outlined in Figure 4. BIRD is a benchmark for evaluating agents' capability to translate a natural language query to a SQL query.

**R.1**. Is fully or at least partially open-sourced.

**Example**: We released the training and validation dataset of BIRD at `https://bird-bench.github.io/`.

**R.2**. Offers an open-source evaluation harness for users.

**Example**: We released the harness to evaluation agents on BIRD at `https://github.com/AlibabaResearch/DAMO-ConvAI/tree/main/bird`.

**R.3**. Includes measures to prevent data contamination, such as a private, held-out test set.

**Example**: We keep a private held-out test set to avoid potential data contamination. Request to evaluate agents on this test set can be submitted at `https://bird-bench.github.io/`.

**R.4**. Includes measures or plans to consistently update challenges over time to avoid overfitting.

**Example**: We plan to consistently update the database and natural language queries to reflect the real-world queries and avoid overfitting. Our updates will be available at `https://bird-bench.github.io/`.

**R.5**. Includes maintenance plans to continuously accept improvements.

**Example**: We will continuously maintain the dataset and evaluation harness, while accepting issues and pull requests from the community at `https://github.com/AlibabaResearch/DAMO-ConvAI/tree/main/bird`.

**R.6**. Clearly states the relationship between the agent capabilities it aims to evaluate and the constructs or outcomes it measures.

**Example**: BIRD evaluates agents' capabilities to serve as a database interface to translate natural language queries into executable SQL queries. To achieve that, BIRD provides agents with a natural language query, the database schema, and SQL-related domain knowledge, and challenges agents to write a SQL query that can be executed to return correct answers.

**R.7**. Clearly states the evaluation subjective of the benchmark (e.g., a model or an agent framework).

**Example**: BIRD is designed to evaluate the capability of ML models as well as the performance of agent frameworks.

**R.8**. Describes steps taken to prevent, identify, and correct flaws.

**Example**: We identify that evaluating generated SQL queries using execution results have two limitations. First, tasks requiring `LIMIT` queries and containing ties in the data may lead to non-deterministic execution results. Second, manually annotated ground-truth queries may contain errors. To understand and mitigate these errors, we randomly sample 500 tasks to perform an additional phase of verification. After verifying queries, we found 11.65% of ground-truth queries are incorrect.[4]

**R.9**. Includes qualitative discussions of the potential impact of unavoidable flaws.

**Example**: The identified incorrect ground-truth queries and potentially more incorrect ground-truth queries in the test dataset can lead to estimation errors of the agent performance and incorrect rankings of agents.

**R.10**. Includes quantitative analysis to assess the impact of unavoidable flaws (e.g., noise of ground truth).

**Example**: We build our quantitative analysis based on the normality assumption. Specifically, suppose the number of data in the test set $N$ is large enough such that the true success rate ($p$) of an agent follows a normal distribution with mean $\mu$ and standard deviation $\sigma$. Given the ground truth's incorrectness rate of $e$ and the estimated agent success rate $p_0$ (based on the imperfect ground truth), $\mu$ and $\sigma$ are calculated as

$$\mu = e + (1 - 2e)p_0; \quad \sigma^2 = \mu(1 - \mu) = (e + (1 - 2e)p_0)\left(1 - e - (1 - 2e)p_0\right)$$

---

[4] We used results by Arcwise [7].

Hence, based on the normality assumption, we can derive a two-sided confidence interval with confidence $\alpha$ for $p$ as follows:

$$\mathbb{P}\left[\mu - 1.96 \times \frac{\sigma}{\sqrt{N}} \leq p \leq \mu + 1.96 \times \frac{\sigma}{\sqrt{N}}\right] \geq 95\% \tag{3}$$

Finally, based on the plug-in estimate (11.65%) for the ground truth's incorrectness rate, we calculate the confidence interval for the agents' performance in Table 16.

**R.11**. Reports metrics about statistical significance, such as confidence intervals.

**Example**: In additional to accuracy estimate, we also calculate confidence intervals for each model in Table 16.

**R.12**. Provides guidance on interpreting results with eval flaws.

**Example**: Given the potential flaws in BIRD, we do not recommend users to rely on the success rate alone for decision-making or selecting models. Instead, we suggest using the confidence interval of the success rate as a reference.

**R.13**. Reports results of non-AI baselines (e.g., human experts).

**Example**: We measured the performance of a SQL expert on BIRD, obtaining a success rate of 92.96%.

**R.14**. Reports results of trivial agents (e.g., one that does nothing).

**Example**: We performed sanity check on our evaluation harness by measuring the performance of a trivial agent that does nothing. We find that the trivial agent achieves 0% success rate, confirming the rigor of our evaluation implementation.

Table 16: Modified Leaderboards of BIRD [38] with Confidence Intervals.

| Method | Dev. Accuracy (%) | Confidence Interval | Original Rank | Possible Rank |
|---|---|---|---|---|
| CHASE-SQL + Gemini | 74.9 | [66.8, 71.4] | 1 | 1-13 |
| Contextual-SQL | 73.5 | [65.7, 70.4] | 2 | 1-16 |
| XiYan-SQL | 73.3 | [65.6, 70.2] | 3 | 1-18 |
| ExSL + granite-34b-code | 72.4 | [64.9, 69.6] | 4 | 1-22 |
| Reasoning-SQL-14B | 72.3 | [64.7, 69.4] | 5 | 1-22 |
| Insights AI | 72.2 | [64.6, 69.4] | 6 | 1-22 |
| TC-SQL | 70.9 | [63.7, 68.4] | 7 | 1-27 |
| Infly-RL-SQL-32B | 70.1 | [63.0, 67.8] | 8 | 1-29 |
| Queryosity | 69.4 | [62.5, 67.3] | 9 | 1-32 |
| OpenSearch-SQL-v2 + GPT-4o | 69.3 | [62.4, 67.2] | 10 | 1-32 |
| GenaSQL | 69.2 | [62.4, 67.2] | 11 | 1-33 |
| OmniSQL-32B | 69.2 | [62.4, 67.1] | 12 | 1-33 |
| OmniSQL-7B | 69.0 | [62.2, 67.0] | 13 | 1-33 |
| PB-SQL + GPT-4o | 68.6 | [61.9, 66.7] | 14 | 2-34 |
| PURPLE + RED + GPT-4o | 68.1 | [61.5, 66.3] | 15 | 2-34 |
| Arcwise + GPT-4o | 68.0 | [61.4, 66.2] | 16 | 2-34 |
| Distillery + GPT-4o | 67.2 | [60.8, 65.6] | 17 | 3-36 |
| RSL-SQL + GPT-4o | 67.2 | [60.8, 65.6] | 18 | 3-36 |
| XiYanSQL-QwenCoder-32B | 67.0 | [60.6, 65.5] | 19 | 4-36 |
| RECAP + Gemini | 67.0 | [60.6, 65.4] | 20 | 4-36 |
| GSR | 66.9 | [60.5, 65.4] | 21 | 4-36 |
| MSL-SQL + DeepSeek-V2.5 | 66.8 | [60.5, 65.3] | 22 | 4-36 |
| AskData + GPT-4o | 65.9 | [59.8, 64.6] | 23 | 7-37 |
| E-SQL + GPT-4o | 65.6 | [59.5, 64.4] | 24 | 7-37 |

Table 16: Modified Leaderboards of BIRD [38] with Confidence Intervals. (Continued)

| | | | | |
|---|---|---|---|---|
| ByteBrain | 65.5 | [59.4, 64.3] | 25 | 7-37 |
| CHESS | 65.0 | [59.1, 63.9] | 26 | 7-37 |
| SCL-SQL | 64.7 | [58.9, 63.7] | 27 | 7-39 |
| EBA-SQL + GPT-4 | 64.6 | [58.8, 63.6] | 28 | 8-39 |
| OeSQL-0.1-Qe-32B | 64.6 | [58.8, 63.6] | 29 | 8-39 |
| RSL-SQL + DeepSeek-v2 | 63.6 | [58.0, 62.8] | 30 | 9-42 |
| Command-A | 63.5 | [57.9, 62.8] | 31 | 9-42 |
| MCS-SQL + GPT-4 | 63.4 | [57.8, 62.7] | 32 | 9-42 |
| PURPLE + GPT-4o | 63.0 | [57.5, 62.4] | 33 | 11-42 |
| GRA-SQL | 62.6 | [57.2, 62.1] | 34 | 14-44 |
| E-SQL + GPT-4o mini | 61.6 | [56.4, 61.4] | 35 | 17-46 |
| OpenSearch-SQL-v1 + GPT-4 | 61.3 | [56.2, 61.2] | 36 | 17-46 |
| Dubo-SQL-v1 | 59.7 | [55.0, 59.9] | 37 | 23-49 |
| SuperSQL | 58.5 | [54.0, 59.0] | 38 | 27-49 |
| SFT CodeS-15B | 58.5 | [54.0, 59.0] | 39 | 27-49 |
| Chat2Query (GPT-4 + data entity modeling) | 58.1 | [53.8, 58.7] | 40 | 30-50 |
| MAC-SQL + GPT-4 | 57.6 | [53.3, 58.3] | 41 | 30-50 |
| SFT CodeS-7B | 57.2 | [53.0, 58.0] | 42 | 30-51 |
| TA-SQL + GPT-4 | 56.2 | [52.3, 57.2] | 43 | 34-51 |
| DeepSeek | 56.1 | [52.2, 57.2] | 44 | 34-51 |
| DTS-SQL + DeepSeek-7B | 55.8 | [52.0, 56.9] | 45 | 35-51 |
| SEE | 55.5 | [51.7, 56.7] | 46 | 35-51 |
| DAIL-SQL + GPT-4 | 54.8 | [51.2, 56.1] | 47 | 37-51 |
| Interactive-T2S | 54.6 | [51.0, 56.0] | 48 | 37-51 |
| Mistral | 53.5 | [50.2, 55.2] | 49 | 37-51 |
| ExSL + granite-20b-code | 51.7 | [48.8, 53.8] | 50 | 40-52 |
| DIN-SQL + GPT-4 | 50.7 | [48.0, 53.1] | 51 | 42-52 |
| GPT-4 | 46.4 | [44.7, 49.7] | 52 | 50-53 |
| Claude-2 | 42.7 | [41.9, 46.9] | 53 | 52-54 |
| Open-SQL | 37.7 | [38.1, 43.0] | 54 | 53-54 |
| Palm-2 | 27.4 | [30.3, 35.0] | 55 | 55-58 |
| ChatGPT + CoT | 25.9 | [29.2, 33.8] | 56 | 55-58 |
| Codex | 25.4 | [28.8, 33.5] | 57 | 55-58 |
| ChatGPT | 24.1 | [27.8, 32.4] | 58 | 55-58 |
| T5-3B | 10.4 | [17.6, 21.6] | 59 | 59-61 |
| T5-Large | 9.7 | [17.1, 21.1] | 60 | 59-61 |
| T5-Base | 6.3 | [14.6, 18.4] | 61 | 59-61 |

