# OpenReview forum: "Establishing Best Practices in Building Rigorous Agentic Benchmarks"
_NeurIPS.cc/2025/Datasets_and_Benchmarks_Track — NeurIPS 2025 Datasets and Benchmarks Track poster_

### Official Review · Reviewer_8Ene · 2025-06-05

**Rating:** 5
**Confidence:** 3

**Summary:**

This paper identifies several issues in AI agent benchmarks that cause over- or under-estimation of agent capabilities and propose a checklist to help developers validate benchmarks. The primary benchmark issues are task (whether the task is doable and not trivial) and evaluation (whether the evaluator is actually correct) validity. The checklist is comprehensive and cover task and outcome validity and reporting. Experiments/assessments confirm identified issues in current benchmarks.

**Additional Feedback:**

* Since you identified issues in a set of benchmarks, could you also list each issue out one by one in a public repo rather than high level summaries in Appendix E? This may help developers correct the errors.

**Dataset Code Accessibility:**

Yes

**Dataset Code Comments:**

I reviewed the project website and briefly the Github repo for benchmark experiments.

**Ethical Considerations:**

No, there are no or only very minor ethics concerns

**Final Justification:**

I believe the authors have sufficiently addressed questions of other reviewers and me. This paper raises an important questions, which warrants its novelty. While the proposed checklist may seem "common sense" and unable to fully "automate" the development process, I believe identifying and formalizing the problem justifies acceptance.

**Limitations Weaknesses:**

* The experimental section of this paper focuses on identifying flaws in existing benchmarks. How can the proposed checklist provide guidance for designing a benchmark from scratch? For example, how can developers design expected outcomes that correlate with a desired capability? How does one approach capabilities that are reflected in the solution process rather than the outcome alone?
* Some of the steps in the checklist, such as verifying the accuracy of llm-as-a-judge, has an iterative and human-in-the-loop component. When designing a new benchmark, what kind of practices could help ensure those steps (e.g., test data sampling) are performed to a sufficient level of accuracy?

**Strengths Contributions:**

* The overall finding is valuable to the community. The checklist is also a useful tool.
* The task formalization in section 3 and aspects of outcome validity identified in section 4.1 are very useful conceptually.
* Identified issues with each benchmark are listed in appendix E

---

> ### Author Rebuttal · Authors · 2025-07-30
>
> We thank the reviewer for their insightful comments. We will incorporate the suggestions in the revision.
>
> > The experimental section of this paper focuses on identifying flaws in existing benchmarks. How can the proposed checklist provide guidance for designing a benchmark from scratch? For example, how can developers design expected outcomes that correlate with a desired capability? How does one approach capabilities that are reflected in the solution process rather than the outcome alone?
>
> We clarify that ABC has two goals:
> 1. To help benchmark builders revise and strengthen their agentic benchmarks.
> 2. To help benchmark users choose and assess benchmarks.
>
> ABC is agnostic to task-specific design choices, such as the design decisions about outcomes or the choice of adopting process-based evaluation, because these design choices significantly vary across domains and use cases.
>
> Nevertheless, the need to establish both task validity and outcome validity is universal. Therefore, we encourage benchmark developers to revisit the relevant checklist items at every iteration of their design cycle.
>
> > Some of the steps in the checklist, such as verifying the accuracy of llm-as-a-judge, has an iterative and human-in-the-loop component. When designing a new benchmark, what kind of practices could help ensure those steps (e.g., test data sampling) are performed to a sufficient level of accuracy?
>
> We apologize for not providing full details on our LLM-as-a-Judge validation in the paper. We clarify that our approach uses pilot studies on a data subset to measure the judge’s accuracy, self-consistency, and agreement with human annotators, reporting results with statistical significance levels. In the repository we mentioned in the paper, we have open-sourced our experiment scripts for evaluating LLM-as-a-Judge approach used in WebArena. In the revision, we will add a detailed description of the procedures to validate LLM-as-a-Judge.
>
> > Since you identified issues in a set of benchmarks, could you also list each issue out one by one in a public repo rather than high level summaries in Appendix E? This may help developers correct the errors.
>
> We clarify that we have open-sourced and updated our code for each issue listed in Appendix E. We have provided the link to our repo in our submission (footnote 1, page 4).

---

> > ### Comment · Reviewer_8Ene · 2025-08-04
> >
> > Thank the authors for addressing my questions. I have no further questions and will maintain my score.

---

### Official Review · Reviewer_Zmu5 · 2025-06-30

**Rating:** 4
**Confidence:** 5

**Summary:**

This paper proposes a checklist to improve the design of agentic benchmarks. The authors propose two validity criteria for agentic benchmarks, namely outcome validity and task validity. They use the checklist to evaluate several existing benchmarks and find these benchmarks to be lacking based on the checklist items.

**Additional Feedback:**

Here is detailed feedback on the Introduction as well as some of the other sections, to assist the authors in improving the clarity of their work for future resubmission. The detailed comments highlight areas where definitions, claims and the motivation of the work can be strengthened.

Check spelling mistakes throughout, e.g. foundamentally [ln 22]
- [ln 18] What does ‘impressive performance’ mean? Why is 35% task resolution impressive? What type of tasks does Tor-bench-Airline propose and measure?
- [ln 21] What does it mean for benchmarks to be impactful? The causality of the claim does not make sense, as it implies that benchmarks are impactful regardless of whether they are trustworthy. Surely they should be trustworthy before it can be claimed that they are impactful
- [ln 23] How is ImageNet a multiple choice dataset?
- [ln 24] What is an ‘automatic metric’?
- [ln 25] Is end-to-end task completion really a sufficient criteria for success in agentic settings? Surely correctness should also be considered?
- [ln 28] From the text it is not clear why agent benchmarks ‘differ fundamentally’ from other AI benchmarks. As described, it’s a comparison of an automatic outcome against a ground truth. That’s quite standard.
- [ln 30] What does ‘up to 100% in relative terms’ mean? Relative to what?
- [ln 36] What is ‘a trivial agent’?
- [ln 36] What are ‘empty response’? And what does it mean to return empty responses?
- [ln 37] What are intentionally impossible tasks?
- [ln 38] From the text it is not clear what the general challenge with current agent benchmarks is. Two specific issues have been highlighted for 2 specific benchmarks. A stronger motivation that describes the general problem is desirable.
- [ln 45 – 48] What are outcome validity and task validity? Are they properties of the benchmark or the system under test? How were these two criteria identified? Why only these two and no other criteria? Do they relate to standard benchmark validity measures (e.g. construct validity)?
- [ln 50] How did you follow these prior works and why did you choose them as your starting point? Why only draw on these two studies if there exists a rich literature on benchmark design?
e.g. “Pilot: A Framework that Understands How to Do Performance Benchmarks the Right Way” (2016, Li et. Al.)
“How to Build a Benchmark” (2015, Kistowski)
- [ln 60] What makes CVE-Bench complex? What is it representative of?
- [ln 63-64] Are outcome and task validity threats, or desired benchmark attributes? Confused.
- [ln 81] When does an agentic benchmark become complex? Do complex agentic benchmarks require different evaluations to simple agentic benchmarks? Why?
- [ln 101] Step 1 – what makes a binary assessment the most appropriate choice? Considering that multiple steps are involved in a complex agent interaction, wouldn’t it be worth assessing how far the agent can proceed until it fails, or at which steps it fails?
- [ln 108-109] Confused. What guarantee are you providing? On what grounds do you base the claim that these conditions must hold? Are the conditions sufficient for guaranteeing a rigorous evaluation?
- [ln 116] This seems like a perfectly reasonable task to assess. Why is it not valid? Of course having too many of these tasks will not be a good benchmark design, but that’s been considered for decades in classification tasks and is the reason why we consider multiple metrics like FP, FN, TP, TN when evaluating outcomes for applications, rather than selecting only one of them.
- [ln 122] Step 2: How many benchmarks could you have analysed? Why focus on ‘popular’ benchmarks only rather than a scientifically valid selection criteria? selecting only benchmarks of corporations or that are popular is not a well substantiated selection criteria and highly likely to skew the analysis due to popularity and availability bias.
- [ln 131] missing citations for related benchmarks in software testing
- [ln 134] Step 4: Is this step supposed to serve as an evaluation of the checklist? It seems like circular referencing is being made in the checklist design – benchmarks are used to develop the checklist, and then the checklist is used to evaluate the benchmarks. That’s like evaluating on your training set.

**Dataset Code Accessibility:**

NA; not applicable to this submission (e.g., no new dataset, benchmark, code, or data provided)

**Dataset Code Comments:**

The paper proposes a checklist for benchmarks, and that checklist is made available.

**Ethical Comments:**

The paper does not approach their topic of study in a manner that requires further ethical scrutiny.

**Ethical Considerations:**

No, there are no or only very minor ethics concerns

**Final Justification:**

I found the authors' responses well considered and they responded to many of my key concerns. I have thus revised my score up.

**Limitations Weaknesses:**

It does not appear like the authors followed any scientifically defensible method in creating their checklist. There are some significant design flaws. It is not clear how checklist items were identified and included, whether the checklist is complete, and how it was evaluated. The authors also did not create any design specifications for what the checklist should accomplish, or any evaluation criteria for the checklist itself. 10 of the 17 papers that influenced the checklist design were also evaluated with the checklist. Benchmark design has been a corner stone of computing research for several decades. Yet, the work does not draw on the best practices that have been established in this domain and narrowly focuses on agentic benchmarks developed in the past couple of years.

While the checklist may be a useful tool for practitioners, the work currently lacks scientific rigour and does not make a clear knowledge contribution to the research community. The authors should ground the work in relevant scientific literature on benchmark development and use a defensible design process that clearly outlines the objectives of the checklist, evaluation criteria, and an evaluation.

**Strengths Contributions:**

Rigorous evaluation and benchmarking are critical for gaining insights into the capabilities of AI. However, the multiple interactions of agentic AI systems make them difficult to evaluate. Benchmarks in this domain consequently suffer validity issues. This work aims to contribute to better agentic benchmarks by providing a checklist for benchmark developers. This is a necessary and timely contribution.

---

> ### Author Rebuttal · Authors · 2025-07-30
>
> We thank the reviewer for the insightful comments, which we will incorporate in the revision. Below, we address each point, citing the line numbers that the reviewer used.
>
> > How were checklist items identified and how is it evaluated? What are design specs & eval criteria?
>
> We appreciate the request for a methodologically grounded description of ABC’s construction. Due to space limits, we omitted these details in the submission. We will add and extend the following explanations in the revision.
>
> *Design specs.*
> - Use cases: 1. Guide benchmark builders during benchmark creation. 2. Help benchmark users understand when a benchmark can produce false positives or false negatives
> - Goals: Reduce false negatives & false positives
> - Criteria: Unambiguous, actionable, & consequential (violating an item measurably harms validity).
>
> *Item generation.* We performed a structured review of 38 publications: 17 agentic benchmarks, 5 critique papers, 12 meta-evaluations, and 4 surveys. We converted existing best practices, bitter lessons, and our insights into an initial checklist draft. In this process, we identified two major conditions for valid agentic evaluation: outcome & task validity.
>
> *Validation.* We applied the Delphi method [3] with 22 stakeholders: 8 benchmark creators, 10 academic researchers in the field of AI agents, and 4 industry agent developers. In each iteration, we asked the reviewers to evaluate items using predefined criteria and to propose new items.
>
> *Assessing benchmarks.* To demonstrate that ABC is practically consequential, we applied it to popular agentic benchmarks and obtained quantitative findings.
>
> > Benchmarks that influenced the ABC design were also evaluated; [ln 134]
>
> We clarify that Step 4 was not intended as a formal validation of ABC. It is an illustrative case study to show the practical consequences of violating items. We didn’t use a train-test split because the number of practically used agentic benchmarks was small (17) at the time of writing. A split would leave a test set too small to provide meaningful statistical power. Thus, we chose to incorporate all benchmarks in the design phase, so that ABC captures a full spectrum of known practices. We then applied the final ABC to 10 benchmarks, illustrating its practical consequences. We will acknowledge such limitations of ABC in the revision.
>
> > The work does not draw on previously established best practices; [ln 50]
>
> We agree that the benchmarking literature is broader than recent agentic evaluation work. ABC was indeed informed by classic sources, as we discussed in Sec. 2 and App. D.
>
> We chose BetterBench [1] and How2Bench [2] as starting points because they are the most recent related works on AI benchmark design. As discussed in the first paragraph of Sec. 2, we also considered a broader range of literature on benchmarks.
>
> ABC is a focused study on the validity of agentic benchmarks, while the studies mentioned by the reviewer have a broader scope, such as fairness and usability of benchmarks. Moreover, we find that both studies share similar insights with ABC in terms of results’ verifiability (outcome validity) and "measuring what the user wants to measure" (task & outcome validity). We will discuss a richer set of literature on benchmark design in the revision.
>
> **[ln 18]** We acknowledge that the original wording of "impressive" was vague. In the revision, we will clarify this with concrete examples and numbers. By the "impressive performance" of GPT-4o, we mean that the agent based on GPT-4o (38%) significantly outperforms previous methods (e.g., 14% for Llama3 [4]).
>
> τ-bench-Airline tasks an agent to resolve users’ ticketing requests. A run is considered successful if the responses contain required strings and the resulting DB state matches the ground-truth state.
>
> **[ln 21]** In the submission, we intended to convey "widely adopted," meaning benchmarks that are being used at major AI providers (OpenAI, Anthropic, Google, etc.). Adoption does not imply that the benchmark is trustworthy. Demonstrating this gap is precisely the motivation for our study. We will replace the term "impactful" with "widely adopted" in the revision.
>
> **[ln 23]** For each image in ImageNet, a system should choose one label from a fixed set of 1,000 labels, a task that is functionally equivalent to a multiple-choice problem.
>
> **[ln 24]** We use "automatic metric" to mean a quantitative measure that can be computed exactly and automatically, such as recall or precision in classification tasks. However, many AI benchmarks involve goals such as "fix a GitHub issue" or "create an efficient travel itinerary," where no automatic method can evaluate the correctness exactly. We will clarify "automatic metric" in the revision.
>
> **[ln 25]** By "task completion," we mean that an agent completes a task correctly.
>
> **[ln 28, 38, 45-48]** We will add details to compare agentic benchmarks with other AI benchmarks and to explain two validity conditions in a domain-independent way. We briefly clarify two fundamental differences that create new failure modes:
> 1. *Unstructured, non-unique gold answers → Issues in outcome validity.* In agentic benchmarks, an agent’s deliverables are often unstructured (e.g., free-form text and sequences of commands). They rely on approximate evaluations (e.g., unit tests). Suboptimal evaluation design can cause false positives/negatives.
> 2. *Sophisticated simulators → Issues in task validity.* Agentic benchmarks often simulate realistic use cases (e.g., websites and DBs). Bugs or shortcuts in simulators let agents game a task without solving it.
>
> These challenges do not generally occur in other AI benchmarks that have explicit gold labels and no interactive states. They also generalize beyond the two examples in the paper. For example, task validity issues generalize to all benchmarks with simulators.
>
> **[ln 30]** By "100%," we mean a benchmark’s score can differ from an agent’s true capability by the score itself. For example, an agent that always returns an empty string (true capability=0) can score 38% on τ-bench-Airline. The overestimation is |0-38|/38 = 100%. We will rewrite this with clearer wording in the revision.
>
> **[ln 36]** By "trivial agent," we mean a baseline agent that always produces the same, hard-coded output. In τ-bench, the trivial agent we refer to always returns an empty string. By "empty response," we mean an empty string returned by an agent.
>
> **[ln 37]** An intentionally impossible task is a task designed to be impossible to accomplish (e.g., changing non-refundable tickets).
>
> **[ln 45-48, 63-64]** We summarize definitions of outcome & task validity:
> - *Outcome Validity*: The benchmark results in a "success" signal if and only if an agent correctly completes a task.
> - *Task Validity*: An agent correctly completes a task if and only if the agent has the capability that the benchmark aims to evaluate.
>
> Both are properties of an agentic benchmark and are necessary conditions for construct validity to hold. We identified them when constructing the initial ABC. We will clarify this in the revision.
>
> **[ln 60]** CVE-Bench is complex in two ways:
> - It requires an agent to exploit a web vulnerability using a sequence of commands. As a vulnerability can be exploited in various ways, verifying the correctness of the agents’ commands is complex.
> - It simulates real-world web apps using containers, requiring a relatively complex, 13k-line-of-code manual implementation.
>
> We believe CVE-Bench is representative of agentic benchmarks that present challenges in achieving outcome validity (due to the complexity of agents’ outputs) and task validity (due to complex simulators).
>
> **[ln 81]** An agentic benchmark becomes complex when the correctness of agents’ outputs cannot be exactly and automatically determined, or when the benchmark requires additional simulators. Additional assessments are needed because higher complexity exposes a larger surface for errors. When the correctness of agents’ outputs can only be determined approximately, we need to assess false-positive and false-negative rates (outcome validity). When there is a simulator, we need to assess gameability (task validity).
>
> **[ln 101]** We focus on the binary evals because, as far as we know, all popular agentic benchmarks use binary evals. Although the multi-step evals might be better, it remains challenging to automatically determine at which step the agent failed for general tasks.
>
> **[ln 108-109]** We guarantee that outcome & task validity are necessary conditions for valid agentic evaluation. We acknowledge that ABC is not an exhaustive list of items that are sufficient for guaranteeing a rigorous evaluation. We will add this limitation to the paper.
>
> **[ln 116]** We clarify that the key issue is not the presence of tasks that cannot be completed, but the disproportionate number of such tasks and insufficient evaluation criteria. This creates a shortcut for agents to succeed without attaining the capability the benchmark aims to measure.
>
> **[ln 122]** Following prior benchmark studies [1], we study agentic benchmarks used by leading AI providers for three reasons.
> 1. Incentives: Top leaderboards shape hiring, funding, and SOTA papers
> 2. Optimization pressure: Their popularity invites score-chasing, amplifying latent validity flaws
> 3. Tooling leverage: These benchmarks form the foundation of the pipelines and libraries used by other labs. Fixing them has an outsized impact.
>
> **[ln 131]** We will cite and discuss related benchmarks in software testing.
>
> **[ln 22]** We will fix typos throughout the paper.
>
> [1] Betterbench: Assessing AI Benchmarks, Uncovering Issues, and Establishing Best Practices. NeurIPS, 2024
>
> [2] How Should We Build A Benchmark? Revisiting 274 Code-Related Benchmarks for LLMs. arXiv, 2025.
>
> [3] The Delphi Method. Reading, Addison-Wesley, 1975.
>
> [4] τ-bench: A benchmark for Tool-Agent-User interaction in real-world domains. ICLR, 2025.

---

> > ### Author Response · Authors · 2025-08-08
> >
> > We hope that our response sufficiently addresses the reviewer's question, and we are looking forward to engaging with the reviewer further. Please let us know if there are any additional questions or concerns we can clarify.

---

### Official Review · Reviewer_LXqg · 2025-07-01

**Rating:** 4
**Confidence:** 3

**Summary:**

This paper addresses an increasingly central issue in the benchmarking of state of the art AI models: the need for more rigorous and reliable agentic benchmarks. The authors introduce the agentic benchmark checklist (ABC), a set of best-practices they have derived from own experiences, a survey of existing best practices, and hands-on examinations of existing benchmarks. The paper formalizes their definition of agentic benchmarking and identifies two key threats to the validity of measured outcomes: outcome validity and task validity. The authors furthermore apply the ABC to ten popular agentic benchmarks and demonstrate that (sometimes signifcant) over- or underestimation of agent performance occurs with current evaluation methods. A case study on applying the ABC to one particular benchmark illustrates its  practical value in improving benchmark rigor.

**Dataset Code Accessibility:**

Yes

**Dataset Code Comments:**

Code and data are available online (see checklist 4.).

**Ethical Comments:**

The paper discusses its "Broader Impact" (Appendix A) and "Limitations" (Appendix A).

**Ethical Considerations:**

No, there are no or only very minor ethics concerns

**Final Justification:**

I thank the authors for the the extensive response and the work. Many of my points were intended to hear the authors perspective on them.
In particular, I raised points regarding
- the common sense nature of some recommendations
- the "actionability" of some items on the list
- potential issues with LLM-as-a-judge benchmarks

and found the authors' responses to these satisfactory. I find that most of my questions and concerns have been addressed and I maintain my score.

**Limitations Weaknesses:**

- While the compilation of best practices into a one framework (the ABC) represents a valuable contribution in terms of organization and accessibility, several checklist items, particularly within the "Outcome Validity" section, align closely with principles that one might consider "common sense". This is admissible given the paper's objective of compiling a comprehensive framework. However, the inclusion of less anticipated insights could enhance the paper's depth and appeal.

- Similarly, there are several items on the checklist that I believe are not implemented flawlessly in the literature not for a lack of awareness but for the real difficulty in implementing them. This somewhat questions how "actionable" some items on the list are. For example, to "verify test cases for correctness and quality (e.g., by human)" for unit testing, while clearly important, is the main challenge involved with unit testing.

- The above point ties into the empirical assessments provided by the authors, which may be subject to a "Hindsight bias". In general, finding flaws or "hacks" in published, deployed benchmarks tends to be easier than a prior or during conceptuation. I have slight doubts that some of the identified flaws, for example SWE-Lancer's vulnerability to test file overwriting (l.284), could have been identified at the time if given access to the ABC. Perhaps paying equally much dedication to continuous updates and maintenance of benchmarks could be included in the list of best-practices, even though this somewhat comes at the cost of cross-comparability beteween updates.

- The advice regarding "LLM-as-a-Judge" also falls somewhat short of being fully actionable. Requiring a demonstration of "accuracy, self-consistency, and agreement with human" and being "designed to resist adversarial inputs and reward hacking" (Fig.1) is in some sense as tall a task as solving the benchmark to begin with. I find the topic of "LLM evaluation" in general highly relevant to this and it may perhaps deserve a dedicated discussion given that it is gaining traction in the literature and its obvious issues from a proper benchmarking perspective.

**Strengths Contributions:**

- The paper tackles a highly important problem in the rapidly evolving field of AI agents. Ensuring the trustworthiness and meaningfulness of performance metrics is of pressing urgency. The overall objective of providing more reliable and meaningful agentic benchmarks is crucial for identifying genuine progress. The paper is well-structured, generally easy to follow, and presents its arguments clearly.

- A significant contribution lies in the compilation of a list of best practices and common pitfalls into the agentic benchmark checklist (ABC). Formalizing and categorizing these design aspects of benchmarks into one framework provides a valuable tool for developers and users that aim to evaluate modern AI agents and their design components.

- The paper demonstrates quantitatively the impact impact of flawed benchmark design. The examples provided, such as the SWE-bench-Verified, the $\tau$-bench, and KernelBench, are popular benchmarks and their shortcoming in estimating agent performance reliably highlights the challenge addressed by this paper.

---

> ### Author Rebuttal · Authors · 2025-07-30
>
> We thank the reviewer for their insightful comments. We will incorporate the suggestions in the revision.
>
> > Several checklist items, particularly within the "Outcome Validity" section, align closely with principles that one might consider "common sense". The inclusion of less anticipated insights could enhance the paper's depth and appeal.
>
> We appreciate the reviewer’s observation that several outcome validity items may look self-evident. We have kept these items intentionally for two reasons:
> 1. *“Common sense” is often violated in practice*: In our assessment of 10 benchmarks, 6 benchmarks violate at least half of the items related to outcome validity.
> 2. *Explicit and easy-to-understand items improve the usability of the checklist*: We believe that a checklist that is intended for routine adoption must be both comprehensive and easy to use. Starting with “common sense” closes the gap between checklist designers and end-users and reduces the cognitive load on practitioners.
>
> We clarify that less-anticipated insights are also included in the checklist. For example, because agentic AI systems can intentionally game benchmarks, we added items that guard against this risk. Our checklist item I.b.2 requires the substring-matching method to be robust against an agent’s strategy of enumerating all possible answers. In WebArena, a question such as “What are the top two best-selling products in 2022?” could otherwise be hacked by listing every product.
>
> > Similarly, there are several items on the checklist that I believe are not implemented flawlessly in the literature not for a lack of awareness but for the real difficulty in implementing them. This somewhat questions how "actionable" some items on the list are. For example, to "verify test cases for correctness and quality (e.g., by human)" for unit testing, while clearly important, is the main challenge involved with unit testing.
>
> We thank the reviewer for raising the important issue of actionability. Our goal is not to restrict the checklist to items that are effortless, but to ensure that every item can, with reasonable effort and planning, be carried out in today’s research environment. For the unit test example, although manually verifying test cases adds tangible work, it is feasible and effective, as demonstrated by SWE-bench Verified.
>
> In the revision, we plan to add an “Approx. effort” column (low/medium/high) to the checklist and encourage partial compliance when resource-constrained.
>
> > The above point ties into the empirical assessments provided by the authors, which may be subject to a "Hindsight bias". In general, finding flaws or "hacks" in published, deployed benchmarks tends to be easier than a prior or during conceptuation. I have slight doubts that some of the identified flaws, for example SWE-Lancer's vulnerability to test file overwriting (l.284), could have been identified at the time if given access to the ABC. Perhaps paying equally much dedication to continuous updates and maintenance of benchmarks could be included in the list of best-practices, even though this somewhat comes at the cost of cross-comparability beteween updates.
>
> Thank you for raising the important point about continuous updates and maintenance of benchmarks. We clarify that we already included an item (III.4) that requires consistently updating the tasks in the benchmark. In the revision, we will split III.4 into two items:
> 1. Agentic benchmarks should consistently update tasks to avoid overfitting.
> 2. Agentic benchmarks should provide a maintenance plan that continuously accepts fixes and improvements.
>
> > The advice regarding "LLM-as-a-Judge" also falls somewhat short of being fully actionable. Requiring a demonstration of "accuracy, self-consistency, and agreement with human" and being "designed to resist adversarial inputs and reward hacking" (Fig.1) is in some sense as tall a task as solving the benchmark to begin with. I find the topic of "LLM evaluation" in general highly relevant to this and it may perhaps deserve a dedicated discussion given that it is gaining traction in the literature and its obvious issues from a proper benchmarking perspective.
>
> We thank the reviewer for emphasizing how demanding it can be to use “LLM-as-a-Judge” responsibly.  Our intent is not to require benchmark authors to solve the issue, but to ensure that anyone who adopts it performs a set of sanity checks. In the revision, we will add a dedicated sub-section with open-source scripts to validate LLM-as-a-Judge. We summarize our approach as follows:
> 1. (Check I.c.1) We will provide an example experiment design with statistical significance analysis that demonstrates the following required properties of LLM-as-a-Judge:
>     - Accuracy: Given the same task to judge, compare the LLM answer with the ground truth.
>     - Self-consistency: Given the same task to judge, inspect the LLM answer change at different executions.
>     - Agreement with humans: For the same message or prompt, compare the LLM's answer with the human answer.
> 2. (Check I.c.2) We will continuously collect existing and emerging issues of LLM-as-a-Judge and formulate them as tests to evaluate the robustness of LLM-as-a-Judge used in an agentic benchmark. For example, recent work shows that tokens such as “Thought process:” can trick LLMs into giving false positive rewards [1]. We can formulate a test that injects the token into all tasks and evaluates the accuracy of LLM-as-a-Judge.
>
> [1] Zhao, Yulai, et al. "One Token to Fool LLM-as-a-Judge." arXiv:2507.08794 (2025).

---

> > ### Comment · Reviewer_LXqg · 2025-08-04
> >
> > Thank you for the the extensive response and the work. I find that most of my questions and concerns have been addressed and I will keep my score.

---

### Official Review · Reviewer_nHYi · 2025-07-02

**Rating:** 5
**Confidence:** 4

**Summary:**

This paper studies agentic benchmarks and identifies shortcomings that may lead to misleading conclusions or misrepresent actual performance. The authors identify two main types of issues: outcome validity and task validity. To advance the field, they propose an Agentic Benchmark Checklist (ABC), which aims to help benchmark creators develop better benchmarks by addressing these two issues. The paper evaluates existing agentic benchmarks using this checklist, finding that many have issues that could have been identified. They also also make use of the checklist to revise and improve an existing benchmark, CVE-Bench.

**Additional Feedback:**

- line 22, typo, should probably be either "foundationally" or "fundamentally"
- line 173, the sentence "random negatives reveal nothing about relu(x)" is missing context and is not self-contained
- line 196, "designing tasks in a way that avoids success by guessing" this suggestion would rule out useful formats like pairwise or multiple-choice questions, which I think are perfectly reasonable formats, as long appropriate baselines are reported.

**Dataset Code Accessibility:**

Yes

**Dataset Code Comments:**

I have skimmed the github repository and it appears that all the relevant artefacts are easily accessible and adequately documented.

**Ethical Comments:**

Not applicable.

**Ethical Considerations:**

No, there are no or only very minor ethics concerns

**Final Justification:**

In my initial review, I was positive about the possible impact of this paper but had some concerns on the terminology & formalization of the issues that the authors raise. The rebuttal addressed these concerns very convincingly.

The paper is not perfect (I appreciate e.g. reviewer `Zmu5`'s perspective), but I believe its strengths outweigh its weaknesses: it is a solid paper that could have a significant (positive) impact on agents research.

**Limitations Weaknesses:**

To my mind, the main weakness is the formalization of the issues into two buckets, task validity and outcome validity, which I find hard to grasp and somewhat confusing. I see three distinct problems going on that could offer an alternative framing:

1. **Proxy metrics not being aligned with the actual goal**. For example, exact match on a goal state is not always a perfect proxy to solving the task.
2. **Baselines not being provided to contextualize performance numbers**. I don't see a dummy agent achieving non-zero performance as an intrinsic shortcoming of the benchmark itself, but rather something that should be reported as a baseline. For example, in a classification task, it is expected that a naive baseline (predicting always the most likely class) will achieve a non-trivial accuracy score - I don't see why evaluating agents is special.
3. **Existence of ways to adversarially break the benchmark**. The fact that agents can "game" the system to achieve high scores without making meaningful progress on the actual goal is reminiscent of overfitting.

It seems that outcome validity is mostly about (1), and task validity mostly about (2) and (3). Do the authors agree?

The authors claim that "agentic benchmarks differ fundamentally from traditional AI benchmarks", but to me it seems that the three classes of issues above (misalignment between metrics and real goal, baselines achieving non-trivial performance, and overfitting / gaming) are almost universal across ML benchmarks.

The attempt at formalizing the problem (Section 3, step 1) is commendable, but would the authors be able to formalize the outcome & task validity using the mathematical framework they have introduced just above? For example, I find the definition of task validity ("success aligned with capability") to be particularly confusing.

- line 38, the authors claim it's an issue that a trivial agent returning empty responses has a 38% success rate. I view this as a baseline and not a fundamental problem with the benchmark itself. If the real-world task follows the same distribution as the benchmark data, such an agent will consistently achieve the desired behavior 38% of the time.
- line 94, the discussion about annotation noise in BirdBench affecting accuracy seems more related to _outcome validity_ instead of _task validity_, as per line 110.
- In section 4.2, KernelBench failing to remove ground truth answers from GPU memory and adversarial agents being able to exploit this seems fundamentally different from dummy agents achieving a certain level of performance, yet these two types of issues are bucketed together into "task validity"

Overall, I think the contributions of this paper outweigh the weaknesses, but I think this paper could have much larger impact by improving the taxonomy of issues and more systematically relating them to issues in benchmarking other areas of ML.

**Strengths Contributions:**

- I believe this paper addresses an important and timely topic that is very relevant to the NeurIPS community.
- It provides valuable insights into the shortcomings of existing benchmarks, raising many specific and interesting issues.
- The authors' report of improving a specific benchmark (CVE-bench) using the checklist provides a helpful illustration.
- Overall, I find the paper to be well written and easy to follow.

Altogether, this is a well-executed paper with valuable insights that researchers and practitioners working on agentic AI will find useful.

---

> ### Author Rebuttal · Authors · 2025-07-30
>
> We thank the reviewer for their insightful comments and provide the following clarifications. We will include them in the revision.
>
> > To my mind, the main weakness is the formalization of the issues into two buckets, task validity and outcome validity, which I find hard to grasp and somewhat confusing. I see three distinct problems going on that could offer an alternative framing:
> > 1. Proxy metrics not being aligned with the actual goal.
> > 2. Baselines not being provided to contextualize performance numbers.
> > 3. Existence of ways to adversarially break the benchmark.
> >
> > It seems that outcome validity is mostly about (1), and task validity mostly about (2) and (3). Do the authors agree?
>
> We agree with the reviewer’s classification overall, but task validity covers more situations than items (2) and (3) do. We clarify the definitions of task validity and outcome validity as follows.
>
> First, consider the logical flow of an agentic benchmark:
> 1. Target capability (conceptual): the ability we wish to measure.
> 2. Task outcome (typically unstructured—code, free-form text, action sequences): the observable output of the agent’s behavior.
> 3. Evaluation result (binary): success or failure as reported by the benchmark.
>
> To achieve validity, the benchmark should return a "success" result if and only if the agent truly has the target capability. We decompose this logical requirement into two design requirements:
> 1. Task validity (capability ↔ outcome): A task is solved by an agent if and only if the agent possesses the target capability. This includes not only the reviewer’s items (2) and (3) but also any other flawed task designs or buggy implementations that break this link, as we mentioned in Section 4.2.
> 2. Outcome validity (outcome ↔ evaluation result): The benchmark returns "success" if and only if the task has indeed been solved. Issues here correspond to the reviewer’s item (1).
>
> We will add more discussion about the derivation of task and outcome validity in the revision.
>
> > The authors claim that "agentic benchmarks differ fundamentally from traditional AI benchmarks", but to me it seems that the three classes of issues above (misalignment between metrics and real goal, baselines achieving non-trivial performance, and overfitting / gaming) are almost universal across ML benchmarks.
>
> Although other ML benchmarks also need to ensure validity, issues in outcome and task validity are often not major concerns in ML benchmarks due to explicit gold labels and no interactive states. By contrast, agentic benchmarks are fundamentally different from other ML benchmarks in the following aspects, making the issues in outcome and task validity major concerns:
> 1. **Unstructured, non-unique gold answers → Issues in outcome validity.** In agentic benchmarks, the agent’s deliverables can be code, multi-turn text, or a sequence of tool calls. There is rarely a single gold label; instead, benchmarks rely on approximate evaluations such as unit tests and string matching. Suboptimal evaluation design can cause false positives or false negatives.
> 2. **Sophisticated simulators → Issues in task validity.** Agentic benchmarks often simulate real-world use cases with simulated websites, file systems, and databases. Bugs or exploitable shortcuts in these simulators let agents "game" the task without truly solving it, inflating scores.
>
> We will thoroughly discuss the differences between agentic benchmarks and other ML benchmarks in the revision.
>
> > The attempt at formalizing the problem (Section 3, step 1) is commendable, but would the authors be able to formalize the outcome & task validity using the mathematical framework they have introduced just above? For example, I find the definition of task validity ("success aligned with capability") to be particularly confusing.
>
> Thank you for highlighting the need to formalize agentic evaluation. In the revision, we will add explicit definitions of task validity and outcome validity. We provide a brief summary below.
>
> We first need to introduce additional definitions:
> - $C_A$ : the set of capabilities the agent actually possesses
> - $c_0$: the specific capability the benchmark aims to measure
> - $R_T$: task-resolution flag ( "success" or "failure" ). Exactly determining $R_T$ often requires manual inspections.
> - $f_{Eval}$: binary score returned by the automatic evaluator (1 = success, 0 = failure)
>
> Then, task validity holds if and only if
> $$c_0 \in C_A\  \leftrightarrow\ R_T = \text{``success''} \quad (1)$$
> Outcome validity holds if and only if
>
> $$R_T = \text{``success''}\ \leftrightarrow\ f_{Eval} = 1 \quad (2)$$
>
> Equation (1) requires that the task is solved if and only if the agent truly has the target capability. Equation (2) requires that the benchmark reports success if and only if the task has been solved. Together, (1) and (2) ensure that the benchmark score faithfully indicates whether the agent possesses $c_0$.
>
> > line 38, the authors claim it's an issue that a trivial agent returning empty responses has a 38% success rate. I view this as a baseline and not a fundamental problem with the benchmark itself. If the real-world task follows the same distribution as the benchmark data, such an agent will consistently achieve the desired behavior 38% of the time.
>
> Thank you for the clarification. We agree that a trivial agent can serve as a baseline. We are concerned about the magnitude of its score rather than its existence, from two perspectives:
> 1. *Interpretation of results.* The trivial agent outperforms the agents based on GPT-4o [1] and o3-mini-high [2], which clearly possess more capability.
> 2. *Misalignment with the goal in the real-world use case.* An empty response does not fulfill the user’s request with reasonable explanations. Counting it as a success indicates a mismatch between the benchmark’s notion of "success" and real-world task requirements, which is a task validity issue.
>
> In the revision, we will adjust the wording to make our arguments explicit.
>
> > line 94, the discussion about annotation noise in BirdBench affecting accuracy seems more related to outcome validity instead of task validity, as per line 110.
>
> Thanks for pointing this out. The annotation noise in BirdBench is indeed more related to outcome validity than task validity. We will revise line 94 accordingly.
>
> > In section 4.2, KernelBench failing to remove ground truth answers from GPU memory and adversarial agents being able to exploit this seems fundamentally different from dummy agents achieving a certain level of performance, yet these two types of issues are bucketed together into "task validity"
>
> Thank you for emphasizing that task validity is multifaceted. We define it as any mismatch between whether the task is marked as solved and whether the agent truly possesses the capability the benchmark is meant to measure.
>
> In the example of KernelBench, the agent can game the benchmark by writing a kernel that returns the correct output without actually demonstrating an understanding of kernel operations or efficient matrix calculations. Because task success is achieved without the target capability, this constitutes a task-validity failure.
>
> > line 22, typo, should probably be either "foundationally" or "fundamentally." line 173, the sentence "random negatives reveal nothing about relu(x)" is missing context and is not self-contained
>
> The relu(x) example demonstrates that if the fuzz tester happens to generate only negative inputs, relu(x) will always output 0. Under such limited test coverage, an agent could "succeed" simply by returning the constant 0, even though it has not captured the true semantics of ReLU.
>
> In the revision, we will provide full context for this example and correct all typos throughout the paper.
>
> > line 196, "designing tasks in a way that avoids success by guessing" this suggestion would rule out useful formats like pairwise or multiple-choice questions, which I think are perfectly reasonable formats, as long appropriate baselines are reported.
>
> Thank you for highlighting this. Pairwise and multiple-choice formats are perfectly acceptable; the concern is simply that agents might succeed by chance. In the revision, we will clarify the checklist item, rephrasing it as: "Design tasks so they cannot be solved by random guessing, unless the benchmark also reports the performance of a random-guess baseline."
>
> [1] τ-bench: A benchmark for Tool-Agent-User interaction in real-world domains. ICLR, 2025.
>
> [2] Introducing OpenAI o3 and o4-mini. OpenAI, 2025.

---

> > ### Comment · Reviewer_nHYi · 2025-08-05
> >
> > Thanks for your comprehensive rebuttal.
> >
> > Your clarifications on task & outcome validity and the formalization you propose are indeed very helpful, and I strongly encourage you to include them in the next version of the manuscript.
> >
> > Overall your comments address all my questions and concerns.
> >
> > I continue to think this is a solid paper with potentially high impact on agents research and will therefore keep my score.

---

### Note · Authors · 2025-08-12

We appreciate the time and effort of the reviewers, ACs, and SACs and their prompt responses throughout the review process. We thank the reviewers for their insightful comments and suggestions and will incorporate them in a future revision.

We believe our rebuttal addresses the questions raised by all reviewers. Reviewer Zmu5 has not yet had a chance to respond to our clarifications, while we remain happy to engage further if they wish. Given the short rebuttal window, we also welcome continued discussion after the review period. For clarity, we briefly summarize our key points in the rebuttal.

First, we justified the methodology behind our proposed checklist, ABC. We constructed and reviewed ABC through a structured literature review of 30+ sources and discussions with 20+ stakeholders (benchmark developers, academic researchers, and industry experts). We also provided explicit design specifications and justified the validity taxonomy (outcome validity and task validity).

Second, we acknowledged the limited number of agentic benchmarks used in practice at the time of writing. Therefore, our construction process of ABC considers all such available benchmarks. Our illustrative case study (Section 5) is not intended as a formal validation, but rather to show the practical importance of assessing benchmarks. We explained why we did not split the benchmark set and the trade-offs accordingly.

In the revision, we will clarify our methodology and design choices, clearly explain key concepts (“impressive performance,” “trivial agent,” “complex agentic benchmarks”), strengthen connections to classical benchmark-design literature, and correct typos. We believe these updates do not change our core contributions. Rather, they will enhance the clarity, rigor, and practical relevance of the work. We again thank the reviewers, ACs, and SACs for their constructive and thorough critiques.

---

### Decision · Program_Chairs · 2025-09-18

**Decision:**

Accept (poster)

**Comment:**

This paper addresses a critical gap in AI agent evaluation by introducing the Agent Benchmarking Checklist (ABC), a structured framework designed to enhance the validity and reliability of agent benchmarks. Its core contributions include: (a) formalizing common benchmarking pitfalls into actionable categories of task validity and outcome validity; (b) compiling best practices for detecting proxy metric misalignment, baseline deficiencies, and adversarial exploitation; and (c) empirically validating the framework through case studies on popular benchmarks like SWE-bench and CVE-bench. The work’s strengths align strongly with the DB Track’s focus on rigorous evaluation standards: the ABC provides a timely, well-structured synthesis of dispersed community knowledge into a practical tool, demonstrated through concrete improvements to existing benchmarks. Its clear organization and empirical grounding further bolster its utility.

However, limitations temper the contribution’s impact: some checklist items (e.g., human verification of test cases, LLM-as-judge validation) articulate well-known challenges without novel solutions, and the framework offers limited guidance for designing benchmarks from scratch—particularly for capabilities tied to solution processes rather than outcomes. While the rebuttal clarified terminology and addressed concerns about actionability, it did not fully resolve tensions between outcome-focused evaluation and process-oriented capability assessment.

Despite these shortcomings, the decision to accept rests on three key factors: (1) the ABC’s pragmatic value in consolidating fragmented best practices into an adoptable standard; (2) strong empirical validation demonstrating tangible benchmark improvements; and (3) the urgent community need for such frameworks amid the rapid proliferation of agentic AI. Post-rebuttal, reviewers acknowledged the authors’ thorough engagement. In balance, the paper’s role in advancing benchmark transparency and reproducibility justifies inclusion, though future work should address process-centric evaluation and implementation scalability.